# A Mechanistic Approach to Replacing Antibiotics with Natural Products in the Treatment of Bacterial Diarrhea

**DOI:** 10.3390/biom15071045

**Published:** 2025-07-18

**Authors:** Mingbang Wei, Huaizhi Liu, Zhefan Hu, Peixiao Wen, Yourong Ye, Yangzom Chamba, Hongliang Zhang, Peng Shang

**Affiliations:** 1College of Animal Science, Xizang Agriculture and Animal Husbandry University, Linzhi 860000, China; weimingbang@xza.edu.cn (M.W.); 202300201094@stu.xza.edu.cn (H.L.); 202400201098@stu.xza.edu.cn (Z.H.); 202400201092@stu.xza.edu.cn (P.W.); yeyourong@xza.edu.cn (Y.Y.); 202400200135@stu.xza.edu.cn (Y.C.); zhanghongliang@xza.edu.cn (H.Z.); 2Key Laboratory of Tibetan Pig Genetic Improvement and Reproduction Engineering, Linzhi 860000, China; 3Tibetan Pig Science and Technology Courtyard in Nyingchi, Linzhi 860000, China; 4Provincial-Ministerial Collaborative Innovation Center for R&D of Agricultural and Animal Husbandry Resources with Tibetan Characteristics, Linzhi 860000, China

**Keywords:** natural products, antibiotics, diarrhea-causing bacteria, pathogenesis, antibiotic mechanism of action, natural product mechanism of action

## Abstract

Natural products have emerged as potential alternatives to antibiotics in the treatment of bacterial diarrhea, due to their multi-targeting effects, low potential for inducing resistance, and favorable safety profiles. Currently, the search for natural product-based therapies has become an emerging focus in medical research. This growing interest is driven by the increasing awareness that the widespread and irrational use of antibiotics has contributed to the alarming rise in antibiotic-resistant bacterial strains, which in turn diminishes the efficacy of conventional drugs. Among these concerns, the limitations of antibiotics in managing bacterial diarrhea and the potential mechanisms by which natural products exert therapeutic effects are the main focus of this paper. Natural products, containing a wide array of bioactive compounds, can not only directly inhibit the growth of pathogenic bacteria, disrupt bacterial membrane synthesis, and reduce toxin production, but also modulate inflammatory responses, enhance immune function, repair intestinal barriers, and restore gut microbial ecology—highlighting their systemic and multi-targeted therapeutic potential. Therefore, this paper will elaborate on how natural products combat bacterial diarrhea from three aspects: the pathogen and pathogenesis of bacterial diarrhea, natural product-based therapeutic studies, and the underlying mechanisms of action, thereby proposing natural products as viable alternatives to antibiotics.

## 1. Introduction

Diarrhea is a common intestinal pathology characterized by a change in the shape of the stool and an increase in the number of bowel movements relative to a healthy organism. Diarrhea can be classified as infectious diarrhea or non-infectious diarrhea. Infectious diarrhea has a large number of causative pathogens, including four species: bacteria, fungi, viruses, and parasites [1]. Bacteria, a group of organisms that cause diarrhea in the organism, are the most abundant and are distributed in four different families: *Vibrionaceae*, *Enterobacteriaceae*, *Bacillaceae*, and *Helicobacteraceae* [2]. Now, according to the Global Health Organization research study, there are roughly 1.7 billion cases of diarrhea in the world every year, of which the percentage of bacterial diarrhea cases is about 30–50% of the total diarrhea cases. Bacterial diarrhea is a common infectious diarrhea in developing countries, and the morbidity and mortality of bacterial diarrhea are also significantly higher than those caused by other pathogens. In China, the prevalence of bacterial diarrhea is geographically and seasonally related, and relatively speaking, the incidence of bacterial diarrhea is higher in the south than in the north [3]. Among the cases of bacterial diarrhea, children and the elderly, a special group, account for a higher proportion of bacterial diarrhea cases because of their lower immunity [4,5]. When children and the elderly are infected with bacterial diarrhea, their low immunity can lead to a series of serious complications and even death. With the widespread and irrational use of antibiotics in today’s society, the number of drug-resistant causative organisms of bacterial diarrhea has been increasing, and drug resistance has been growing, leading to more difficult means of treating bacterial diarrhea, and threatening the public health safety of the whole society [6]. With this background, the mechanisms of antibiotics and bacterial resistance have become essential to understand before exploring alternatives.

Studies have shown that there are a wide variety of traditional antibiotics used in the treatment of bacterial diarrhea, which can be broadly classified into tetracyclines, sulfonamides, fluoroquinolones, cephalosporins, macrolides, and other antibiotics according to their chemical properties [7]. Their mechanism of action in the clinic is mainly bacteriostatic or bactericidal, and the inhibition or killing of bacteria mainly contains four major mechanisms of action, including inhibition of bacterial cell wall synthesis, enhancement of bacterial cell membrane permeability, interference with bacterial protein synthesis, and inhibition of bacterial nucleic acid replication and transcription and other effects. These drugs have good therapeutic effects in the treatment of diarrhea caused by bacterial infections, especially against *Salmonella*, *E. coli*, *Shigella*, *Campylobacter*, and other common diarrhea-causing bacteria. However, with the extensive use of antibiotics and irrational use of antibiotics in China, resulting in a variety of diarrhea-causing bacteria on antibiotic resistance is becoming more and more serious, and accompanied by antibiotic treatment related complications, in order to solve the related problems, the study of the bacterial drug resistance mechanism is indispensable [8]. The bacterial resistance mechanism of bacterial diarrhea mainly includes four aspects: target site change, antibiotic inactivating enzyme production, drug efflux pump action, and cell membrane permeability change [9]. Many diarrhea-causing bacteria have resistance to sulfonamides and quinolones, and one of the most significant mechanisms is target site alteration. The pharmacological effect of quinolones is mainly through specific binding to the enzyme targets of diarrhea-causing pathogens, resulting in the inhibition of diarrhea-causing bacteria’s ability to generate DNA rotamase and topoisomerase IV, which leads to the inhibition of the process of bacterial DNA self-replication [10]. Diarrhea-causing bacteria such as *Campylobacter*, *Escherichia coli*, and *Shigella* are able to alter the specific binding sites of antibiotic drugs by using genetic mutations in the enzyme sites, resulting in unaffected bacterial DNA self-replication and drug resistance [11]. The pharmacological effects of sulfonamides are mainly on dihydropteroate synthase and dihydrofolate reductase in the folate synthesis pathway, and some diarrhea-causing bacteria are also capable of acquiring resistance by mutating the target enzyme genes [12]. One of the notable mechanisms leading to the emergence of β-lactam (e.g., penicillin, cephalosporin) resistance in diarrhea-causing bacteria is the production of antibiotic-inactivating enzymes. *Escherichia coli* and *Salmonella* produce ultra broad-spectrum β-lactamases (ESBLs) capable of hydrolyzing third-generation cephalosporins (such as ceftriaxone, cefotaxime). In addition, *Salmonella* and *Shigella* produce aminotransferases, which are capable of inactivating aminoglycoside antibiotics, leading to the development of drug resistance [13,14]. Drug efflux pumping is mainly widespread in Gram-negative enteric pathogens that cause diarrhea and is an important factor in their ability to develop multi-drug resistance. *Escherichia coli* and *Campylobacter*, a group of Gram-negative bacteria, express multiple RND-type efflux pumps, which function by being able to pump drugs such as tetracycline and fluoroquinolones out of the cell, leading to a decrease in the optimal drug concentration for the antibiotic treatment of bacteria and a decrease in the therapeutic efficacy of the drugs, resulting in the development of high levels of resistance [15,16]. Altered cell membrane permeability is also one of the important strategies for the development of the drug resistance mechanism in Gram-negative bacteria; the channel proteins (e.g., OmpF, OmpC) on their outer membrane can regulate the entry and exit of antibiotic molecules into and out of the cell, and when these channel proteins are structurally altered or the pore expression is down-regulated, it will lead to a decrease in the efficiency of the entry of antibiotic drugs into the cell, and a decrease in the effectiveness of the therapeutic treatments, thus generating drug resistance [17]. In addition, the long-term use of antibiotics in killing pathogenic bacteria in the intestinal tract usually causes extensive damage to many commensal microorganisms in the intestinal tract, and the end result will lead to a reduction in the diversity of the intestinal flora and intestinal flora dysbiosis. Reduced levels of a range of beneficial bacteria, such as probiotics, may be accompanied by an overproliferation of drug-resistant or opportunistic pathogens (e.g., *Clostridium difficile*, *Candida*) [18]. This disruption of the balance of the organism’s microecosystem not only reduces the barrier function and colonization resistance of the intestinal tract, but also inhibits the production of other substances, such as short-chain fatty acids, thereby affecting the body’s immune homeostasis. Long-term or repeated use of antibiotics may also lead to the horizontal transfer of drug-resistant genes in the intestinal flora, exacerbating the risk of drug resistance and laying the foundation for the development of chronic inflammation, allergy, metabolic syndrome, and other systemic diseases [19]. In the use of antibiotics for the treatment of diarrhea, in addition to the ability of antibiotics to kill pathogenic bacteria to produce a therapeutic effect, the body will also produce a variety of adverse reactions. In terms of gastrointestinal symptoms, adverse reactions include vomiting, nausea, abdominal pain, and pharmacogenetic diarrhea, which are usually caused by dysbiosis of the intestinal flora and overgrowth of opportunistic pathogenic bacteria after antibiotic administration. In terms of the body’s immune system, allergic reactions are also more common, usually manifested as itching, erythema, skin fever, etc., after antibiotic administration, which can cause angioedema and anaphylaxis in the body in severe cases. In terms of the antibiotic drugs themselves, the compound quality of some drugs can directly lead to adverse effects in the organism; fluoroquinolones can cause tendonitis or tendon rupture [20], tetracyclines may cause enamel dysplasia and photosensitization [21], and sulphonamides carry a risk of severe hematotoxicity and renal crystallization, and long or repeated doses may also impair liver and kidney function, trigger neurological symptoms (headache, dizziness, even seizures), and spread drug-resistant genes in the gut through horizontal gene transfer, exacerbating the difficulty of subsequent treatment [22]. In conclusion, the existence of these mechanisms not only makes clinical treatment more difficult but also increases the risk of the spread of drug-resistant strains, which is a serious threat to public health. Therefore, there is an urgent need to develop new therapeutic strategies in today’s society, among which the search for natural products to replace antibiotics in the treatment of bacterial diarrhea is one of the important strategies. Against this backdrop, developing safer and more effective alternatives such as natural products has gained increasing attention.

In nature, there are abundant sources of natural products, mainly from plant, animal, and microbial sources [23]. Common natural plant products include herbs such as Chenpi, Angelica sinensis, and nutmeg, and also include Tibetan herbs such as horehound and Himalayan purple jasmine. Common natural animal-derived products include propolis, bezoar, and snake venom, and natural microbial products include antimicrobial peptides, polysaccharides, enzymes, and secondary metabolites of their pairs produced by fungi and probiotics. Existing studies have shown that natural medicines can be used as alternatives or adjunctive therapeutic products to antibiotic-based drugs. Compared with traditional antibiotic therapeutic drugs, natural drugs have demonstrated several advantages in the treatment of infectious diarrhea caused by bacteria. First, the multi-target mechanism of drugs, as one of the characteristics of the mechanism of natural products for the treatment of bacterial diarrhea; natural products can not only directly enter the body to act on the lesion site, directly kill bacteria or inhibit bacterial activity, but also through the regulation of the host body’s immune system, endocrine system, digestive system, and indirectly to achieve the improvement of the richness of the intestinal bacterial flora, the repair of intestinal protective barriers and other multi-target, synergistic therapeutic effects [24]. Compared with traditional antibiotics, natural medicines have multiple targets and do not have a single dependency, so it is not easy for bacteria to develop drug resistance with the long-term use of natural medicines. Secondly, natural drugs are characterized by high safety and low side effects. In the composition of the human population, children and the elderly are a special group of sick people, whose immunity is relatively low, and the number of clinical treatment cases is high. Many natural products are derived from traditional Chinese medicine, functional plants or natural microbial metabolites, which are more suitable for the treatment of children and the elderly. Third, natural medicines are characterized by multiple active ingredients, and the active ingredients include flavonoids, polyphenols, alkaloids, terpenoids, etc., which have pharmacological effects such as anti-inflammatory, antioxidant, and antibacterial effects [25]. While treating bacterial diarrhea in patients, natural products can directly act on the disease site to kill bacteria or inhibit bacterial growth, and also enhance the host organism’s immunity, maintain the homeostasis of the microecological environment of the intestinal flora, and repair the intestinal protective barriers, etc., which not only achieves the effect of treating diarrhea, but also reduces the risk of preventing diarrhea from reoccurring in the organism [26]. When treating bacterial diarrhea, natural products can be considered to be used in combination with intestinal probiotics as well as low amounts of traditional antibiotics, which can reduce the use of antibiotics as well as reduce the probability of the emergence of drug-resistant bacteria while the synergistic effect of multiple drugs enhances the therapeutic effect. In conclusion, natural products have a great market prospect in the treatment of bacterial diarrhea as an emerging therapeutic strategy of “bacterial control + intestinal protection + bacterial community regulation”, and are also an important development direction for antibiotic replacement products in the subsequent research.

In view of the increasing burden of antibiotic resistance and the clinical challenges of managing bacterial diarrhea, this review aims to comprehensively summarize current knowledge on natural product-based interventions. It highlights their multi-targeted therapeutic mechanisms, safety advantages, and potential for integration into modern anti-infective strategies, thereby providing a scientific basis for the development of antibiotic alternatives.

## 2. Pathogenesis Mechanisms of Bacterial Diarrhea Pathogens and Limitations of the Antibiotic Therapy Mechanism of Action

### 2.1. Pathogenic Mechanisms of Common Bacterial Diarrhea Pathogens

Bacterial diarrhea is an infectious diarrheal disease, and its causative factor is a disease caused by the infection of the gastrointestinal tract by pathogenic bacteria. Its pathogenesis mainly includes the entry of bacteria into the intestinal epithelial cells from the outside by destroying the protective barrier of the intestinal tract, the release of bacterial toxins after colonization, and the induction of inflammatory reactions in the body. Common diarrhea-causing pathogens include *Salmonella*, *Escherichia coli*, *Campylobacter*, *Shigella*, *Clostridium difficile*, and *Clostridium perfringens*, which can be transmitted through a variety of ways. In this section, the main bacteriological characteristics of the above pathogens and their pathogenic mechanisms will be highlighted (Table 1).

*Escherichia coli (E. coli)* is a common intestinal diarrhea-causing pathogen, and its pathogenicity depends on the strain virulence factors, serotypes, and their mechanisms of interaction with the host intestine. The four main types of diarrhea-causing *E. coli* include *enteropathogenic (EPEC)*, *enterotoxigenic (ETEC)*, *enteroinvasive (EIEC)*, and *enterohaemorrhagic (EHEC)* [36]. *EPEC* adheres to the small intestinal epithelium through bacterial hairs and utilizes the type III secretion system to inject effector proteins, which disrupts the structure of the microvilli and causes watery diarrhea in infants [37]. *ETEC* produces heat-stable toxins (STs) and heat-unstable toxins (LTs), which activate intracellular cGMP and cAMP pathways, respectively, and promote intestinal water-electrolyte secretion, and are the main causes of traveler’s diarrhea and children’s diarrhea in developing countries [38]. *EIEC* invades colonic epithelial cells and multiplies within the cytoplasm, inducing mucosal inflammation and pus-blood stools, with a clinical manifestation similar to that of bacillary dysentery [39]. *EHEC* on the other hand causes diarrhea in infants through the production of EHEC, damages the intestinal mucosa and vascular endothelium through the production of Shiga-like toxin (Stx), and in addition to diarrhea, some cases may be complicated by hemolytic uremic syndrome (HUS), which manifests itself as hemolytic anemia, thrombocytopenia, and acute renal failure, and has a high lethality rate [40]. Most of the above strains are transmitted via the fecal-oral route, and contaminated food or drinking water is the main source of infection. *Salmonella* spp. are an important pathogen causing bacterial diarrhea, and their pathogenic mechanism is similar to that of enteropathogenic *Escherichia coli (EPEC)*, which injects virulence factors into the intestinal epithelial cells through the type III secretion system, triggering inflammatory reactions, destroying the intestinal mucosal barrier and interfering with cell signaling, leading to water and electrolyte disorders, and ultimately inducing diarrhea [41]. *Campylobacter* spp., another common foodborne pathogen, have a unique curved morphology and flagellar driving ability, which can effectively penetrate the intestinal mucosal barrier. Its secreted heat-stable enterotoxin can disrupt the structure of epithelial cells, weakening the ability of water absorption and inducing secretory diarrhea [42]. *Shigella* spp. are the main causative agent of bacillary dysentery, and its pathogenicity is mainly dependent on the synergistic effect of strong invasiveness and Shiga toxin (Shiga toxin). The bacterium can enter the colonic epithelium through M-cells and multiply and spread within the cells, causing apoptosis of the epithelial cells and ulceration of the intestinal mucosa, triggering an intense inflammatory response [43]. *Clostridioides difficile*, a Gram-positive anaerobic bacillus, is the main causative agent of antibiotic-associated diarrhea, and its pathogenicity is mainly attributed to the secretion of toxin A (enterotoxin) and toxin B (cytotoxin), which both disrupt the structure of intestinal epithelial cells, causing severe inflammation and mucosal damage, leading to apoptosis and intestinal dysfunction. The bacterium is transmitted through the fecal-oral route, and the spores are highly resistant to the environment, and can be transmitted indirectly through contaminated medical equipment, bed sheets, or the hands of healthcare workers. Prolonged use of antibiotics is one of its important causative factors [44]. *Clostridium perfringens*, a Gram-positive anaerobic bacillus, is an important causative agent of foodborne diarrhea. Type A strains act on small intestinal epithelial cells by secreting enterotoxin (Clostridium perfringens enterotoxin, CPE), which disrupts the permeability of the cell membrane, induces the excessive secretion of water and electrolytes into the intestinal lumen, leading to watery diarrhea. Electrolytes are secreted in large quantities and activate the acute inflammatory response. Infections are most often caused by the ingestion of improperly stored cooked foods, which can lead to outbreaks of food poisoning, especially in group dining settings [45].

### 2.2. Pathogenesis of Bacterial Diarrhea Caused by Intestinal Flora Dysbiosis

Intestinal dysbiosis is defined as a series of intestinal dysfunctions triggered by the disruption of the homeostasis of the intestinal microbial microenvironment in a normal organism, with a decrease in the number of beneficial bacteria and an increase in the number of harmful bacteria, with diarrhea being one of the main symptoms [46]. When the intestinal flora is dysbiotic, the ratio of beneficial bacteria to pathogenic bacteria is damaged, and the beneficial bacteria are unable to effectively inhibit the pathogenic bacteria, which leads to pathogenic bacteria being more likely to adhere, colonize, and release toxins in the intestinal lumen. In terms of affecting the protective function of the intestinal barrier, intestinal dysbiosis will lead to impaired intestinal barrier function and increase the permeability of the intestinal mucosa, so that pathogenic microorganisms and their metabolites (e.g., lipopolysaccharides, toxin A, toxin B) can pass through the intestinal epithelial cells to induce the degradation of tight junction proteins and activate the innate immune response of the host, which is typically manifested in the activation of the *NF-κB* pathway and the release of a large number of pro-inflammatory factors (such as *IL-1β*, *TNF-α*), leading to damage of intestinal epithelial cells, exacerbation of local inflammation, and ultimately triggering intestinal secretory or exudative diarrhea [47]. In terms of intestinal flora metabolism, intestinal flora serves as one of the important players in regulating intestinal metabolic homeostasis. Normal commensal flora produce short-chain fatty acids (e.g., butyric acid, propionic acid, etc.) through the fermentation of dietary fiber, and these metabolites not only provide energy for intestinal epithelial cells, but also have anti-inflammatory and epithelial repair-promoting effects [48]. When the intestinal flora is dysbiotic, the production of short-chain fatty acids decreases, leading to impeded energy supply to the cells, changes in the acid-base environment in the intestinal tract, and a decrease in the healing ability of the intestinal mucosa, which further contributes to diarrhea in the organism. More importantly, intestinal flora dysbiosis-induced diarrhea is not simply limited to local reactions but may also affect the overall health of the body through systemic inflammation. There are many factors leading to intestinal flora dysbiosis, among which the extensive use and irrational use of antibiotics is one of the main reasons. While targeting the killing of pathogens, antibiotic drugs also produce collateral damage to other commensal bacteria in the intestinal tract, destroying the colonization of these beneficial bacteria in the body, and providing a suitable environment for the growth and reproduction of the conditionally pathogenic bacteria to survive [49]. For example, *Clostridium difficile*, which can form spores in the intestinal environment, is highly tolerant, and its released toxins A and B are highly cytotoxic to the intestinal mucosa, which is a typical causative factor of antibiotic-associated diarrhea. In order to effectively control diarrhea caused by intestinal flora dysbiosis, related therapeutic strategies are more inclined to microecological regulation. Among them, the application of probiotics and synbiotics has been shown to some extent to promote the recovery of beneficial bacterial populations, inhibit pathogenic bacterial colonization, and improve the immune status of the host [50]. In addition, emerging therapies such as fecal transplantation have shown great potential in re-establishing the structure of the intestinal flora, alleviating diarrhea symptoms, and preventing recurrence [51]. Therefore, restoring the balance of intestinal flora, such as the use of probiotics, fecal transplants, dietary modifications, and avoidance of antibiotic abuse, are key measures for the treatment and prevention of such diarrhea [52] (Figure 1).

### 2.3. Mechanisms of Action and Limitations of Antibiotics in the Treatment of Bacterial Diarrhea

The mechanism of antibiotic treatment of bacterial diarrhea is mainly through the inhibition or killing of infection-causing pathogenic bacteria, thus controlling intestinal infection, reducing bacterial toxin production, reducing inflammatory response, and promoting recovery [53]. Different types of antibiotics for the treatment of diarrhea have different mechanisms of action. This paragraph will describe four typical antibiotics for the treatment of bacterial diarrhea (Figure 2). Quinolone antibiotic drugs can block the self-replication and transcription process of bacterial DNA by inhibiting bacterial DNA rotamase and topoisomerase IV, which can fundamentally perform the role of inhibiting the growth and reproduction of bacteria. β-lactams (amoxicillin, cephalosporin) can cause bacterial death by inhibiting the synthesis of the bacterial cell wall, so that the bacterial structure is damaged [54]. Aminoglycosides (e.g., neomycin, gentamicin) and macrolides (e.g., erythromycin) can cause bacterial death by causing damage to bacterial ribosome function and preventing protein synthesis [55]. The therapeutic mechanism of antibiotics guarantees the normalization of all physiological functions in the intestinal tract, effectively reduces the number of pathogenic bacteria in the intestinal tract, and reduces the burden on the intestinal tract. Antibiotics play an important role in the treatment of bacterial diarrhea in terms of controlling bacterial infection and shortening the course of the disease, but there are considerable limitations. First, extensive use or irrational use of antibiotic drugs can lead to dysregulation of the body’s normal intestinal flora, disrupting the microenvironmental homeostasis in the intestinal tract, resulting in a decrease in the number of beneficial bacteria in the intestinal tract, and an excessive increase in the number of pathogenic and opportunistic pathogens (e.g., *Clostridium difficile*), which triggers intestinal antibiotic-associated diarrhea. Second, some studies have shown that some bacteria have now become resistant to antibiotics, such as drug-resistant *Salmonella*, drug-resistant *Shigella*, and drug-resistant *E. coli* strains, and conventional antibiotic treatments are ineffective and more difficult to treat [56]. Third, patients infected with bacterial diarrhea do not necessarily require antibiotic treatment, such as *enterohemorrhagic E. coli (EHEC)*, whose use of antibiotics may promote the release of toxins, interfere with the natural immune process, and increase the risk of complications [57]. Fourth, antibiotics, as therapeutic drug monomers, cannot directly neutralize the problem with the toxins produced by the bacteria after reaching the acting bacteria during treatment, and the therapeutic effect of antibiotics will become extremely limited if and when the causative organism has already produced a large number of toxins prior to antibiotic treatment. Antibiotics have different effects in different organisms and can produce certain adverse reactions, such as allergic reactions, liver and kidney toxicity [58]. In summary, antibiotics as the current mainstream drugs for the treatment of bacterial diarrhea are effective, but the limitations are more obvious. Therefore, in the process of treating bacterial diarrhea, it is necessary to clarify the therapeutic effect of antibiotics and the effect of drug sensitivity experiments in order to use them, and it is also necessary to cooperate with nutrition, replenish body fluids, and regulate the ecological balance of the intestine during the process of medication, so as to make antibiotics achieve the optimal efficacy.

In addition, when treating certain types of bacterial diarrhea caused by antibiotic-induced gut microbiota dysbiosis—such as *Clostridioides difficile* infection (CDI), which is clinically characterized by severe diarrhea, pseudomembranous colitis, and even life-threatening complications—the limitations of current therapies underscore the urgent need for alternatives to broad-spectrum antibiotics. Natural products represent a promising class of such alternatives. Among the wide range of bioactive compounds, several natural substances have shown potential in preventing or alleviating CDI. Polyphenols, such as tea polyphenols and proanthocyanidins from grape seeds, can selectively promote the growth of beneficial bacteria (e.g., Lactobacillus, Bifidobacterium), restore microbial balance, and thereby reduce the risk of *C. difficile* colonization [59]. Flavonoids like quercetin and baicalin not only exhibit antioxidant and anti-inflammatory properties but also help maintain the expression of tight junction proteins, enhance intestinal barrier integrity, and reduce toxin translocation and inflammation [60]. Alkaloids such as berberine have been demonstrated in several animal models to significantly reduce CDI incidence, with mechanisms including inhibition of toxin production, modulation of bile acid metabolism, and promotion of commensal microbial recovery [61]. These advantages of natural products provide a strong basis for their continued investigation as potential alternatives to broad-spectrum antibiotics in the treatment of bacterial diarrhea.

## 3. Progress in the Study of Natural Product Extracts for the Treatment of Bacterial Diarrhea

### 3.1. Main Sources and Types of Natural Products

The sources of natural medicines are very wide and varied, with the main sources including plant sources, animal sources, microbial sources, and their microbial metabolites. Among them, plant sources are the most important components of natural drug sources, such as Chenpi, honeysuckle, nutmeg, and other common herbs, and Tibetan herbs such as Himalayan jasmine and horehound, which are rich in flavonoids, polyphenols, terpenoids, alkaloids, and other active ingredients characterized by their significant antimicrobial, anti-inflammatory, and antioxidant effects. Animal sources are also one of the important components of natural drug sources, of which it has been shown that substances such as propolis, oxalis, snake bile, snake venom, etc., have certain antibacterial effects and immunomodulatory effects [62]. Microbial sources as a component of natural drug sources; in addition to their own ability to produce natural drugs, their metabolites are also one of the components that make up natural drug sources [63]. In nature, some bacteria (actinomycetes), fungi, and probiotics can produce antimicrobial peptides, polysaccharides, enzymes, and secondary metabolites, of which antibiotics such as penicillin and erythromycin first originated from microbial metabolism. In addition, algae and deep-sea microorganisms in the ocean are also rich in biologically active substances, and the deep-sea field has become a new direction in the search for natural drugs [64]. Due to the two characteristics of natural medicines, namely the wide source and the complexity of composition, they provide a material basis and research basis for the multi-target treatment of bacterial diarrhea, as well as a multi-selective therapeutic herb library, and also provide diversified choices for the treatment of bacterial diarrhea with antibiotics.

### 3.2. Representative Natural Products for the Treatment of Bacterial Diarrhea

There is a wide range of natural medicines for the treatment of bacterial diarrhea, in which the active ingredients of the medicines have antibacterial, anti-inflammatory, and antioxidant characteristics, regulating cellular immune activity and repairing the intestinal barrier function. Among them, flavonoids, polyphenols, alkaloids, terpenoids, and organic acids can be used as representative natural products against bacterial diarrhea, which have high medicinal value and market application prospects [65] (Table 2).

Flavonoids are widely found in various herbs and plants, containing bioactive substances such as baicalein, quercetin, and rutin, which have significant antibacterial and anti-inflammatory effects. Among them, baicalein is derived from Scutellaria baicalensis and is one of its main active substance components. It has been shown that baicalein has an inhibitory effect on a variety of pathogenic bacteria, such as Salmonella, Shigella, and Escherichia coli, and can inhibit intestinal inflammatory responses and reduce intestinal mucosal damage through the regulation of signaling pathways, such as *TLR4/NF-κB* (by inhibiting *NF-κB* activation), *MAPK* (by suppressing p38 phosphorylation), and so on [79]. Quercetin, as another typical class of flavonoid bioactive substances with significant antioxidant and immunomodulatory functions, can reduce pathogenic bacteria-induced intestinal epithelial barrier disruption, improve tight junction protein expression, and enhance intestinal barrier function [80]. Alkaloidal natural products are also one of the important research objects for the treatment of bacterial diarrhea, present in traditional Chinese medicines such as Huanglian and Huangbai, represented by berberine, a pharmacologically active substance with broad-spectrum antimicrobial activity, which is able to directly inhibit the growth of *Salmonella*, *Shigella*, and *virulence-producing Escherichia coli*, as well as to effectively block the invasion process of pathogenic bacteria [81]. Berberine can also alleviate the inflammatory response by inhibiting the expression of intestinal epithelial *TLR4* and reducing the release of inflammatory factors such as *IL-1β* and *TNF-α*, and it also shows significant advantages in regulating the intestinal flora [82]. In addition, berberine can enhance the intestinal barrier function and repair the intestinal mucosal structure, and it is one of the more mature natural antimicrobial components in current research and clinical application. Terpenoids, including oleanolic acid, ursolic acid, curcumin and other pharmacologically active substances, play an important role in the treatment of bacterial diarrhea as natural medicines, and these pharmacologically active substances have significant antimicrobial activity, but also by regulating intestinal oxidative stress and immune status, they alleviate the mucosal inflammation caused by bacterial infection [73]. Among them, curcumin can not only inhibit the apoptosis of intestinal epithelium triggered by pathogenic bacteria, but also promote intestinal repair by regulating signaling pathways such as *PI3K/Akt* (through activating the *PI3K/Akt* cascade) and *Nrf2/HO-1* (by stimulating *Nrf2* nuclear translocation and *HO-1* expression) [83,84]. In addition, they show significant effects in regulating intestinal flora and reducing intestinal permeability. Organic acids include short-chain fatty acids, citric acid, malic acid, and other chemically active substances, among which short-chain fatty acids (acetic acid, propionic acid, and butyric acid) can be produced by fermentation of Ran polysaccharides or prebiotics, which have significant antibacterial and anti-inflammatory properties [85]. Acetate is widely produced by bacterial groups such as Bifidobacterium and Bacteroides [86]. With its simple molecular structure, acetate can rapidly cross the intestinal barrier into the bloodstream and be transported to organs such as the liver and muscles, where it plays roles in lipid synthesis, cholesterol metabolism, and appetite regulation. Within the gut, acetate can enhance intestinal epithelial barrier function by activating receptors such as GPR43, promoting the expression of tight junction proteins, and inducing the secretion of mucus and antimicrobial peptides by intestinal epithelial cells, thereby inhibiting the colonization and invasion of pathogenic bacteria [87]. Additionally, acetate can induce the generation of regulatory T (Treg) cells and regulate the Th1/Th2 immune balance, effectively alleviating chronic inflammation caused by dysbiosis [88]. Propionate is primarily produced in the gut through the succinate and propionyl-CoA pathways and serves as a key metabolite linking gut microbiota with the host glucose metabolism [89]. In the liver, propionate can act as a substrate for gluconeogenesis and regulate blood glucose homeostasis. Immunological studies have shown that propionate can enhance the chemotaxis and functional maturation of intestinal immune cells by activating receptors such as GPR41, thereby strengthening the mucosal immune barrier. Moreover, propionate can modulate the inflammatory responses of macrophages and dendritic cells, suppressing the overexpression of pro-inflammatory cytokines such as *IL-6*, *IL-1β*, and *TNF-α*. This contributes to alleviating gut inflammation-related disorders such as inflammatory bowel disease (IBD) and irritable bowel syndrome (IBS) [90]. Although butyrate is present in the smallest amount among the major short-chain fatty acids, it has the most critical physiological functions. Butyrate is the primary energy source for colonic epithelial cells, with approximately 70% of it being directly utilized by these cells to promote proliferation and differentiation, thus maintaining the integrity of the intestinal mucosa [91]. Furthermore, butyrate exhibits significant anti-inflammatory effects by inhibiting histone deacetylase (HDAC) activity, affecting inflammatory signaling pathways such as *NF-κB*, and reducing the expression levels of inflammatory mediators. It also promotes the differentiation of Treg cells and suppresses Th17-related immune responses, thereby maintaining intestinal immune tolerance and reducing the occurrence of autoimmune and chronic inflammatory conditions. Studies have shown that reduced levels of butyrate are closely associated with various intestinal diseases, such as Crohn’s disease, ulcerative colitis, and colorectal cancer. Therefore, butyrate is considered an important indicator for assessing gut health and a potential target for therapeutic intervention in intestinal disorders [92]. Short-chain fatty acids can directly inhibit the growth of a variety of Gram-negative bacilli, while providing energy to intestinal epithelial cells, maintaining their metabolic activity and barrier function, and modulating the immune response through the activation of receptors, such as *GPR41/43*, which helps to reduce the incidence of diarrhea [93]. Polyphenolic compounds, in addition to encompassing flavonoids, are also widely distributed in natural plant sources such as green tea, grape seeds, and goji berries. Representative bioactive substances, including chlorogenic acid, catechins, and epigallocatechin gallate (EGCG), have been shown not only to exert significant antibacterial activity but also to regulate the gut microbiota, thereby contributing to the maintenance of intestinal homeostasis [94]. Among them, epigallocatechin gallate can not only inhibit the release of toxins by Shigella and reduce its adhesion ability on cells, but also regulate the *NF-κB* pathway in intestinal cells (by suppressing *NF-κB* nuclear translocation), thereby reducing the expression of cellular inflammatory factors [71]. Meanwhile, polyphenolic compounds play a role in promoting the growth and development of probiotics and inhibiting the reproduction and expansion of pathogenic bacteria, which play an important role in maintaining the microecological balance of probiotic and pathogenic intestinal microbiota. In conclusion, all of these representative natural medicines have shown significant advantageous effects in antimicrobial aspects, as well as good biocompatibility properties. In terms of immunomodulation, they are able to regulate the relevant pathways to reduce the production of inflammatory factors, while protecting the intestinal mucosa from damage. These natural drugs not only provide the theoretical basis and material guarantee for finding antibiotic drug substitutes, but also provide important support for building green and safe antibacterial strategies.

### 3.3. In Vitro and In Vivo Modeling of Natural Products for the Treatment of Bacterial Diarrhea in Animals

In recent years, some researchers have made great progress in the use of natural products for the treatment of bacterial diarrhea by using in vitro and animal research models. These studies not only validate the potential of natural products for antibacterial, anti-inflammatory, and repair of intestinal function, but also provide a solid foundation for the elucidation of their mechanisms and clinical translation.

In terms of in vitro experiments, natural products have been widely used to evaluate their inhibitory activity against pathogenic bacteria (e.g., *Salmonella typhimurium*, *Shigella*, *pathogenic Escherichia coli*, *Vibrio cholerae*, etc.), and the main assays were determined by the agar diffusion method, minimum inhibitory concentration (MIC), and minimum bactericidal concentration (MBC) test methods [95]. For example, baicalein drug-sensitive paper tablets showed inhibitory effects on a variety of pathogenic bacteria, and the drug-sensitive tablets were attached to Petri dishes containing bacteria, and after incubation for a period of time, obvious inhibitory circles appeared on the Petri dishes, and after MIC and MBC measurements, it was found that the MIC value was significantly lower than that of traditional antibiotics. Berberine not only inhibited bacteria in vitro, but also significantly inhibited the adhesion, invasion, and biofilm formation of pathogenic bacteria. Meanwhile, using intestinal epithelial cell models (e.g., Caco-2 cells, HT-29 cells) to mimic the interactions between pathogenic bacteria and intestinal epithelium, it was found that a variety of natural products could ameliorate the disruption of cellular tight junctions, apoptosis, and over-release of inflammatory factors resulting from pathogenic bacterial infection [96,97]. In addition, the effects of natural products on signaling pathways (e.g., *TLR4/NF-κB*, *MAPK*, *PI3K/Akt*, etc.—including the inhibition of *NF-κB* activation and activation of *PI3K/Akt* phosphorylation) and inflammatory factors (e.g., IL-6, TNF-α, and IL-1β) can be observed by fluorescent staining, Western blot, qPCR, ELISA, etc., to further reveal the mechanism of action of the natural drugs. In terms of animal model research, mice, rats, and rabbits are commonly used as model animals for bacterial diarrhea. Relevant researchers usually induce animals to produce acute diarrhea models by gavage or the intraperitoneal injection of pathogenic bacteria (e.g., *Salmonella typhimurium*, *ETEC E. coli*), which simulates the process of human intestinal bacterial infection. Subsequently, the experimental animals were divided into four groups, usually the control group, the model group, the natural drug treatment group, and the positive drug group, and the weight changes, diarrhea scores, intestinal histopathological changes, and inflammatory marker levels of the animals in each group were observed, and the experimental data in each group were well documented and analyzed using biological statistics. Studies have shown that berberine, curcumin, quercetin, and other natural products have good anti-diarrhea effects in animals, which can significantly reduce diarrhea symptoms, inhibit intestinal inflammation, reduce the content of bacteria in the intestinal tract, and repair the damaged intestinal mucosal structure. For example, quercetin up-regulates intestinal tight junction proteins such as ZO-1, occluding, and claudin-1, and improves intestinal barrier function. Curcumin, on the other hand, alleviates oxidative stress and protects intestinal epithelial cells from damage via the *Nrf2/HO-1* pathway (by activating *Nrf2* and upregulating *HO-1* expression) [98,99]. Notably, some studies have also combined 16S rRNA sequencing technology to confirm that natural products can significantly improve the imbalance of the intestinal flora after infection, increase the abundance of beneficial bacteria, and inhibit the overpopulation of pathogenic bacteria, thus maintaining intestinal microecological homeostasis [100]. Overall, in vitro and animal experiments have shown that natural products have the advantages of multi-targets and low side effects in antibacterial diarrhea, especially in antibacterial and anti-inflammatory activities, repairing the protective barrier of the intestinal tract and regulating the bacterial flora.

## 4. Mechanisms of Action of Natural Products

### 4.1. Antimicrobial and Inhibitory Mechanisms of Natural Products in the Treatment of Bacterial Diarrhea

In the treatment of bacterial-induced diarrhea, natural products have a variety of mechanisms to play an important role in antibacterial, antimicrobial, anti-inflammatory, and antioxidant activities, and the inhibition of bacterial cell membrane synthesis and permeability changes is one of its core mechanisms. Bacterial cell membrane is the structure that maintains the most basic life activities and physiological functions of bacteria, and it is also an important protective barrier for the homeostasis of its internal environment. Many natural active ingredients in natural products can directly target the binding sites on the cell membrane, destroying the functional structure and integrity of its cell membrane, leading to the leakage of cellular contents, and ultimately causing the death of the bacteria [101]. Among the active substances of natural drugs, berberine, flavonoids, and polyphenolic compounds have good lipophilicity, which can be effectively incorporated into the phospholipid bilayer membrane of bacteria, changing the fluidity and stability of its membrane, increasing permeability, causing a large amount of electrolytes, nucleic acids, proteins, and other cellular contents to leak, resulting in a disturbance of the homeostasis of the intracellular environment, and the appearance of a difference in intra- and extracellular osmolality, resulting in the inhibition of bacterial growth or bacterial inactivation [102]. In addition, flavonoid natural products can play an interfering role in the process of bacterial cell wall synthesis. They can affect peptidoglycan synthesis by inhibiting D-alanyl-D-alanine synthetase or binding to peptidoglycan structures, thus inhibiting the normal formation of the bacterial cell wall and leading to easy cell lysis and death [103]. There are also some polyphenolic compounds (e.g., tea polyphenols, eugenol, etc.) in natural products, which can induce peroxidation of lipid substances in bacterial cell membranes, disrupt the structure of the cell membrane phospholipid bilayer and affect the physiological functions of key proteins on the membrane, such as ATP synthase, substance transporter proteins, membrane channel proteins, etc., which can impede the obtaining of energy for the basic life activities of the bacterial energy and the transportation of substances [104]. In addition, natural products can interfere with membrane potential and inhibit proton pump function, leading to cellular energy depletion. It is now found that curcumin is able to prevent bacteria from maintaining the proton gradient and blocking the synthesis of the intracellular energy substance ATP by inhibiting the F_0_F_1_-ATPase activity of *Escherichia coli*, resulting in the inability of energy to satisfy the bacteria’s basic physiological functional activities, which ultimately causes bacterial death [105]. Certain natural antimicrobial peptide compounds can also perforate the bacterial membrane, forming irreversible membrane-penetrating pores, which directly lead to the collapse of the membrane potential and the leakage of the contents, triggering bacterial death [106]. Overall, natural products can directly disrupt the homeostasis of the internal environment of bacteria through multiple pathways, such as interfering with membrane structure, inhibiting membrane lipid synthesis, inducing changes in membrane permeability and membrane dysfunction, to achieve rapid bactericidal and broad-spectrum antibacterial physiological effects. This multi-targeting mechanism is also one of the reasons why natural products are difficult to induce bacterial resistance, which lays an important pharmacological foundation for their use as therapeutic drugs in replacing antibiotics in the treatment of bacterial diarrhea (Figure 3).

Another important mechanism of action of natural products in the treatment of bacterial diarrhea is to interfere with bacterial proliferation and virulence factor expression by inhibiting the synthesis of bacterial proteins and DNA, which is similar to the mechanism of traditional antibiotics. This type of mechanism is mostly found in certain flavonoids, alkaloids, polyphenols, and antimicrobial peptides with special structures. Among them, natural polyphenols such as quercetin and epigallocatechin gallate (EGCG) can bind to bacterial nucleic acids to form a stabilizing complex, thus blocking DNA self-replication and transcription processes [107]. These polyphenolic compound molecules can be inserted into the DNA double helix structure and inhibit the binding activity of DNA polymerase and RNA polymerase, leading to the failure of DNA strand elongation or transcriptional arrest, and in this way, preventing bacteria from generating relevant toxin proteins and structural proteins [108]. Meanwhile, it has been found that some alkaloidal natural products (e.g., berberine) have been shown to interfere with the superhelical structure of bacterial DNA and inhibit the activity of topoisomerases, thus hindering the replication and self-repair of bacterial DNA and achieving an effect similar to that of quinolone antibiotics [109]. In addition, natural products can also act on bacterial ribosomes, interfering with the translation process of generating protein turnover. Some studies have shown that certain flavonoids in natural products can block the binding site of bacterial transporter RNA and inhibit peptide chain elongation reaction by binding to the 30S and 50S ribosomal subunits of bacteria, so as to interrupt the protein synthesis of bacteria. A similar mechanism of action is found in natural antimicrobial peptides, which specifically recognize bacterial ribosomes and, upon specific binding, affect bacterial ribosome translation efficiency. The increased probability of mistranslation of the bacterial ribosome leads to the production of bacterial nonfunctional proteins, which inhibits bacterial growth and development and the expression level of bacterial virulence factors [110]. More importantly, some natural products can also indirectly reduce the synthesis of bacterial virulence factors and structural proteins by downregulating the expression of genes related to pathogenicity of pathogenic bacteria (e.g., *toxin genes*, *adhesin genes*, *flagellar synthesis genes*, etc.), which can effectively alleviate the degree of diarrhea caused by bacterial infection [111]. Overall, these natural products can fundamentally inhibit bacterial reproduction and pathogenicity by disrupting the synthesis of bacterial genetic material and regulating the expression pathway of virulence factors, and their multi-targeting and low induced resistance advantages will make them important alternatives for the development of new antimicrobial drugs and the replacement of antibiotics (Figure 4).

In the treatment of bacterial diarrhea, natural products can attenuate the pathogenicity of pathogenic bacteria by inhibiting their toxin synthesis and secretion in addition to the direct inhibition of bacterial growth, especially in interfering with key virulence factors such as the type III secretion system (T3SS), heat-stable enterotoxin (ST), and heat-unstable enterotoxin (LT), etc. The T3SS is a Gram-negative protein injection device unique to pathogenic bacteria (e.g., *Salmonella*, *Shigella*, *pathogenic E. coli*, etc.), which injects the toxin proteins produced by the bacteria directly into the host cells, inducing the reorganization of the basic cellular skeleton, leading to apoptosis and the production of inflammatory responses, and the whole process is also one of the core mechanisms of bacterial colonization and invasion [112]. Nowadays, it has been found that a variety of natural products, such as baicalein, curcumin, and chlorogenic acid, can inhibit the structural assembly and toxin secretion of the T3SS by interfering with the expression of T3SS-related genes (e.g., *hilA*, *invF*, and *sipA*, etc.) or the function of their encoded proteins [113]. For example, curcumin down-regulated the expression of SipA/SipB proteins in *Salmonella* T3SS and significantly reduced its invasive ability on host cells. On the other hand, against LT and ST enterotoxins secreted by virulence-producing *Escherichia coli (ETEC)*, natural products can also effectively inhibit their expression or course of action. For example, quercetin and berberine can significantly reduce the expression of genes encoding LT (eltAB) and ST (estA) toxins in *ETEC*, thus reducing the synthesis and release of toxin proteins, decreasing their ability to activate the cAMP and cGMP signaling pathways of the host cells, and attenuating diarrhea caused by excessive secretion of intestinal fluid [114]. In addition, some of the natural polyphenolic compounds can directly or indirectly contact with already secreted enterotoxins, blocking their binding process to host receptors and further reducing toxin activity [115]. This mechanism of natural drugs directly targeting the action of bacterial toxins rather than the bacteria themselves is characterized by the fact that it does not pose a direct threat to the survival of the bacteria, thus making it more difficult to induce bacterial resistance, and at the same time avoids disrupting the ratio of the intestinal flora. In conclusion, by inhibiting the synthesis, secretion and function of toxins such as T3SS, LT, ST, etc., natural products provide a new strategy for the treatment of bacterial diseases that “does not disrupt the structure of the intestinal flora but reduces the virulence of the bacteria”. It provides a safe, efficient, and low-risk of drug resistance alternative for the treatment of bacterial diarrhea (Figure 5).

### 4.2. Anti-Inflammatory Mechanisms of Natural Products in the Treatment of Bacterial Diarrhea

In the pathogenesis of bacterial diarrhea, intestinal epithelial cells infected by pathogenic bacteria will rapidly activate the host immune system, releasing a large number of pro-inflammatory factors, such as *IL-1β*, *TNF-α*, and *IL-6*, etc., and these cytokines can lead to inflammatory cascade reaction, intestinal barrier disruption, tissue edema, and increased secretion, thus aggravating diarrhea symptoms [116,117]. The significant advantage of natural products in anti-inflammation is one of the important mechanisms for their treatment of bacterial diarrhea. Nowadays, studies have shown that a variety of natural products (e.g., curcumin, quercetin, berberine, chlorogenic acid, etc.) have good anti-inflammatory activity. The mechanism is mainly by inhibiting the expression and release of inflammatory factors, thus blocking the activation of inflammatory signaling pathways [118]. Curcumin can inhibit the transcription of downstream pro-inflammatory factor genes by blocking the *TLR4/MyD88/NF-κB* signaling axis, thereby significantly reducing the mRNA and protein expression levels of *IL-1β*, *TNF-α*, and *IL-6* [119]. Berberine can affect the activation process of inflammatory factors in the nucleus and reduce cytokine synthesis by inhibiting the phosphorylation of p38 and JNK in the *MAPK* pathway [120]. Some of the natural products also up-regulate the expression of anti-inflammatory factors, regulate the balance of immune response, and attenuate the tissue damage brought about by the inflammatory response. Quercetin has been shown to significantly inhibit *NF-κB* activity in intestinal epithelial cells induced by pathogenic bacteria, which in turn inhibits the expression of a variety of inflammatory factors and protects the intestinal barrier structure [121]. Some natural polyphenolic compounds can also exert antioxidant and anti-inflammatory effects by activating the *Nrf2/HO-1* pathway, indirectly modulating the production of inflammatory factors and preventing oxidative stress-induced cell damage [122]. In animal experiments, the expression levels of inflammatory factors in serum and in intestinal tissues were significantly decreased in the treatment group treated with a variety of natural products relative to the model infection group, and after HE staining, the intestinal tissue structure was well restored, and the inflammatory cell infiltration was reduced, which further determined the anti-inflammatory effect of natural products in the organism [123]. In summary, natural products can effectively reduce the intestinal inflammatory response triggered by bacterial infection by regulating the expression of inflammatory factors, blocking the activation of inflammatory pathways, and restoring the inflammatory balance, and play an important immunomodulatory and intestinal protective role in the treatment of bacterial diarrhea.

### 4.3. Mechanism of Action of Natural Products in the Treatment of Bacterial Diarrhea by Maintaining the Protective Barrier of the Intestine

One of the key mechanisms of natural medicine for the treatment of bacterial diarrhea is the maintenance of the intestinal barrier function. The intestinal protective barrier consists of intestinal epithelial cells, tight junction proteins (e.g., ZO-1, occludin, claudin-1), the intestinal immune system, and the intestinal flora, which are composed of four components. The protective intestinal barrier is the body’s first important line of defense against invasion by pathogenic bacteria, preventing penetration of toxins, and maintaining intestinal homeostasis [124]. During bacterial infections, pathogenic bacteria and the toxins they secrete disrupt the structure and function of the intestinal epithelial cells, decrease the expression of tight junction proteins, and lead to an increase in the permeability of the intestinal epithelial barrier, which triggers leakage of the intestinal epithelial cell contents and inflammatory responses [125]. Natural products can protect and repair the damaged intestinal barrier through multiple mechanisms of action. At the cellular structure level, a variety of natural products (e.g., quercetin, chlorogenic acid, curcumin, berberine, etc.) can up-regulate the expression of tight junction proteins, which enhances the tightness of the junctions between the epithelial cells and prevents various toxins produced by bacteria from entering the epithelial cells through the epithelial barrier and causing cellular damage. Curcumin can enhance the expression of ZO-1 protein and occludin protein by activating the *AMPK* signaling pathway and inhibiting the *NF-κB* signaling pathway, thus attenuating LPS-induced intestinal barrier damage. Natural products also reduce pathogenic bacteria adhesion and invasion, indirectly protecting intestinal epithelial cells from destruction [75]. Certain plant polysaccharides and saponin-like components also stimulate intestinal mucus secretion and enhance the physical barrier function of the intestine. In addition to this, natural products have antioxidant effects, which can scavenge the large amount of reactive oxygen species (ROS) produced by bacteria during infection of the host organism and alleviate the damage caused to intestinal cells because of oxidative stress. Quercetin can prevent apoptosis of intestinal epithelial cells by activating the *Nrf2/HO-1* pathway and enhancing cellular antioxidant capacity [126]. In terms of the microenvironment of intestinal flora, natural products can not only maintain the ecological balance of the intestinal microenvironment and inhibit the proliferation of harmful bacteria, but also promote the growth of beneficial bacteria (such as *Bifidobacteria* and *Lactobacillus*), to achieve the purpose of maintaining the stability of the intestinal barrier environment [127]. Animal experiments proved that the mice in the treatment group given natural product treatment, after HE staining, had significantly reduced intestinal tissue damage compared with the model group, and their tight junction protein expression content returned to normal after treatment, and the inflammation of intestinal tissues and intestinal flora dysbiosis were significantly improved [128]. Overall, the synergistic effect of natural products in protecting the tight junction structure of intestinal tissues, anti-inflammatory and antioxidant, and regulating the intestinal flora in several aspects led to the protection of the integrity of the intestinal barrier, and also played an important therapeutic role for natural products in preventing bacterial infections and dissemination (Figure 6).

### 4.4. Mechanism of Action of Natural Products in the Treatment of Bacterial Diarrhea by Regulating the Structure of the Intestinal Flora

Natural products in the course of the treatment of bacterial diarrhea are accompanied by adjustments in the structure of the intestinal flora in the organism. When the organism has diarrhea after bacterial infection, it is usually accompanied by a decrease in the number of beneficial bacteria in the intestinal tract and an increase in the number of harmful bacteria, the homeostasis of the intestinal internal environment is disrupted, and the intestinal protective barrier is also damaged [129]. Regulating the structure of intestinal flora as one of the important mechanisms of natural products in the treatment of bacterial diarrhea, natural products can not only help the body to maintain a healthy intestinal flora internal environment homeostasis, but also maintain the intestinal barrier integrity. Natural products also have physiological functions such as the competitive inhibition of pathogenic bacteria colonization, regulation of immune response, and promotion of nutrient metabolism. Active substances such as polysaccharides, flavonoids, polyphenols, saponins, and other active ingredients contained in natural products have significant advantages in regulating the ability of intestinal flora. For one thing, the natural products themselves can serve as metabolic substances for beneficial bacteria in the intestinal flora, providing life-essential substances for beneficial bacteria and promoting the growth and reproduction of beneficial bacteria [130]. Among them, polysaccharides include astragalus polysaccharides, lycium polysaccharides, etc., which can be directly utilized by probiotic *bifidobacteria* and *lactobacilli*, and after a series of life metabolic activities, the generation of short-chain fatty acids by these probiotic bacteria will be promoted, which ensures the acid-base balance of the intestinal environment, and its short-chain fatty acids provide an acidic environment that can inhibit the growth and reproduction of harmful bacteria [131]. In addition to polysaccharides, various naturally derived bioactive compounds—such as polyphenols, flavonoids, alkaloids, and terpenoids—can also modulate the gut microbiota composition through multiple mechanisms, thereby influencing SCFA production and maintaining intestinal homeostasis. For instance, polyphenolic compounds like tea polyphenols and proanthocyanidins from grape seeds are poorly absorbed in the colon and can be metabolized by specific microbial species. This process promotes the growth of beneficial bacteria (e.g., Bacteroides, Lactobacillus, and Bifidobacterium) while inhibiting opportunistic pathogens, thus optimizing microbial composition [132]. Certain flavonoids can further increase the abundance of butyrate-producing bacteria, elevating overall SCFA levels and enhancing mucosal barrier function and anti-inflammatory responses [133]. Alkaloids such as berberine have been shown to indirectly regulate microbial metabolic activity via the AMPK signaling pathway, leading to the increased production of propionate and butyrate [134]. Therefore, these bioactive compounds interact with gut microbiota not only to improve the microbial ecosystem but also to boost SCFA generation, exerting anti-inflammatory, immunoregulatory, and barrier-repairing effects—making them important natural modulators for maintaining and restoring intestinal health. Secondly, some natural products also have the ability to kill bacteria directly, and can directly act on specific harmful bacteria (such as *Escherichia coli*, *Salmonella*, *Shigella*, etc.), and play a killing or inhibiting effect on these bacteria, while the life activities of probiotics in the whole process are less affected, which is conducive to the restoration of intestinal flora dysbiosis to normal [135]. Third, natural products are closely related to the regulation of the immune system of the body’s internal environment. Natural products can regulate the intestinal flora to maintain homeostasis by influencing the body’s immune regulatory system, and quercetin and berberine can reduce the level of intestinal inflammation, improve the environment of intestinal mucosal microflora, and promote the growth and reproduction of probiotics in the intestinal tract [136,137]. It is now shown that in an induction model of infected mice with bacterial diarrhea, natural products were found to significantly improve the abundance of intestinal flora in the intestinal contents through therapeutic effects, with an increase in the proportions of Bacteroidetes phylum and Thick-walled Bacteroidetes phylum, an increase in the number of beneficial bacteria, and a decrease in the number of harmful bacteria in the intestinal flora of infected mice [138]. It has also been shown that natural products can enhance the adhesion of intestinal epithelial cells to specific beneficial bacteria and assist in the establishment of stable symbiotic relationships [139]. In conclusion, natural products can assist in the therapeutic process through a variety of pathways such as nutritional support, selective bacterial inhibition, improvement of the microenvironment and regulation of immunity, which together assist in restoring the structure and function of intestinal flora, and fundamentally solving the problem of intestinal health, and exerting the therapeutic role of natural products in many ways.

### 4.5. Immunomodulatory Mechanisms of Natural Products in the Treatment of Bacterial Diarrhea

The immunomodulatory mechanism of natural products in the treatment of bacterial diarrhea is an important part of their comprehensive therapeutic effect. Bacterial infection triggers local and systemic immune responses in the intestinal mucosa, which can aggravate tissue damage and diarrhea symptoms if the immune response is ineffective or multidimensional [140]. Natural products can restore the body’s immune homeostasis by regulating innate and acquired immune functions, thereby reducing the pathological damage caused by bacterial infection. In terms of innate immune regulation, berberine, curcumin, and chlorogenic acid can effectively inhibit the activation of Toll-like receptors (*TLRs*), especially *TLR4*, and block their downstream *MyD88/NF-κB* and *MAPK* signaling pathways, thereby reducing the release of pro-inflammatory factors (e.g., *TNF-α*, *IL-1β*, *IL-6*) and decreasing intestinal damage caused by immune over-activation [141]. Meanwhile, natural products can also enhance the expression of anti-inflammatory factors, such as *IL-10*, and promote the transformation of macrophages from M1-type to M2-type, realizing the shift in immune regulation from pro-inflammatory to anti-inflammatory [142]. In terms of acquired immunomodulation, natural products have the function of influencing the proliferation and differentiation of T cells. Quercetin and certain saponin bioactives can reduce Th17 cell-mediated inflammatory responses by regulating the balance of Th1/Th2 and Th17/Treg, while enhancing the negative regulation of immunity by Treg cells [143]. In addition, natural polysaccharides have been shown to stimulate the production of IgA antibodies in gut-associated lymphoid tissue (GALT), enhance local mucosal immune defenses in the gut, and act as a barrier to invading pathogenic bacteria and toxins. Some studies have also shown that natural products can maintain the normal function of immune cells by activating the *Nrf2* signaling pathway and attenuating reactive oxygen species (ROS)-induced apoptosis and dysfunction of immune cells [144]. Now, in the model of mice with bacterial-induced diarrhea, the mice treated with natural products, relative to the model group of mice that were not treated after bacterial infection, it was found that the splenic organ index, lymphocyte activity, and pro-inflammatory factors in the mice in the treated group were significantly lower than those of the model group, and at the same time, the content of the antibody level in the treated group also increased significantly, which also shows the role of natural products in the treatment of the high antibody levels and macrophage phagocytosis during the course of bacterial diarrhea, suggesting that it has an important impact on treating the inflammation [145]. In summary, natural products can make multiple immune cells and signaling pathways coordinate with each other, so as to effectively restore the balance of immunoregulation in the intestinal environment, which is conducive to the body to avoid the syndrome reaction and tissue damage caused by the imbalance of immunoregulation, so the natural products in the treatment of bacterial diarrhea play an important role in immunoregulation (Figure 7).

## 5. Challenges and Future Research Directions

There are notable limitations in current research on the mechanisms by which natural product extracts may replace antibiotics in treating bacterial diarrhea. Most existing studies rely on preclinical animal models, which differ significantly from human biology in terms of immune response, gut microbiota composition, and metabolism. These differences reduce the translational value of animal data and limit the application of findings to human health. Although natural products have demonstrated advantages such as multi-target effects, low potential for resistance, and overall safety, several critical challenges hinder their widespread and rational clinical application.

First, the chemical complexity of natural products—with diverse structural types and multiple bioactive compounds—makes it difficult to isolate the specific active constituents and clearly define their mechanisms of action. Most pharmacological effects arise from the synergistic interaction of these compounds, which also complicates quality control and reproducibility. Many studies still rely on crude extracts or mixtures, lacking purification and quantitative assessment of key active ingredients, which affects batch consistency and therapeutic predictability. Second, bioavailability remains a major bottleneck. Compounds such as polyphenols and flavonoids are often unstable or poorly absorbed when administered orally, as they are rapidly metabolized or degraded in the gastrointestinal tract. This limits the ability to maintain effective drug concentrations at the target site, ultimately compromising clinical efficacy. Novel drug delivery technologies—such as targeted release systems or nanocarriers—are needed to improve absorption and ensure therapeutic levels in vivo. Third, the “slow-acting” nature of many natural antibacterial agents reduces their suitability for acute infections compared to conventional antibiotics, which act rapidly. A potential solution is to explore synergistic treatment strategies, combining natural products with low doses of antibiotics to improve efficacy while minimizing resistance development. Fourth, the lack of large-scale, well-controlled human clinical trials hinders the clinical translation of natural product-based therapies. Most current data are derived from in vitro studies or small animal models, leaving safety, efficacy, dosing, and indications in humans inadequately defined. Fifth, standardization is a major challenge. The composition of natural extracts can vary due to differences in plant origin, harvesting season, and extraction techniques. This variability affects both quality assurance and regulatory approval. Moreover, unsustainable or unethical sourcing—such as overharvesting of endangered species—raises ecological and ethical concerns.

In the future, rigorous human clinical studies must be prioritized, alongside the development of advanced extraction and purification technologies to ensure reproducibility and clarity in mechanisms. Innovative drug delivery platforms should be employed to enhance bioavailability. Sustainable production should be supported through artificial cultivation, fermentation, or the chemical synthesis of bioactive compounds to ensure scalable, ethical sourcing. Furthermore, the integration of multi-omics technologies (e.g., transcriptomics, metabolomics, microbiomics) and computational tools (e.g., network pharmacology) can help elucidate interactions along the host–microbiota–immunity axis. In summary, while natural products present a promising strategy for replacing antibiotics in bacterial diarrhea treatment, their advancement into mainstream clinical application requires overcoming significant scientific, technical, and ethical challenges. Future progress will depend on multidisciplinary collaboration, technological innovation, and policy support to fully realize their therapeutic potential.

## 6. Conclusions

In conclusion, natural products have demonstrated multiple advantages in the program as an alternative to antibiotics in the treatment of bacterial diarrhea, which not only have the ability to directly inhibit the growth and development of pathogenic bacteria, but also achieve the modulation of the body’s immune response, repair of the intestinal barrier structure, repair of the intestinal barrier structure, and reconstruction of intestinal flora homeostasis, by destroying the structure of the cell membrane of pathogenic bacteria, interfering with the synthesis of bacterial proteins and DNA, inhibiting the secretion of bacterial toxins, and decreasing the expression of the body’s inflammatory factors and rebuilding the homeostasis of intestinal flora, which are also the multi-target mechanisms of natural products in the treatment of the bacterial diarrhea disease process. These mechanisms reflect the holistic, synergistic, and mild nature of natural products as therapeutic drugs in the therapeutic process and also alleviate the drug resistance problem caused by the abuse of antibiotics compared with traditional antibiotics. However, natural products still face challenges in terms of complexity of composition, low bioavailability, unknown mechanism of action and insufficient clinical validation. Future research should focus on clarifying the effective drug active ingredients in natural products, optimizing the research methods of natural product extracts, designing natural product targeted delivery systems, and improving the targeting and clinical drug utilization efficiency of natural products with the help of modern pharmacology, bioinformatics, and synthetic biotechnology. At the same time, it is necessary to standardize the establishment of the standard of herbs from which natural products are derived and the evaluation system of clinical medicinal effects, as well as to speed up the construction of the experimental research model from animal to human conversion, which are the keys to promoting the natural products from the laboratory to the clinical application. In conclusion, natural products in the treatment of bacterial diarrhea multi-mechanism, multi-target characteristics have broad research prospects, and are expected to become an important auxiliary program or even alternative strategies to antibiotics in the future.

## Figures and Tables

**Figure 1 biomolecules-15-01045-f001:**
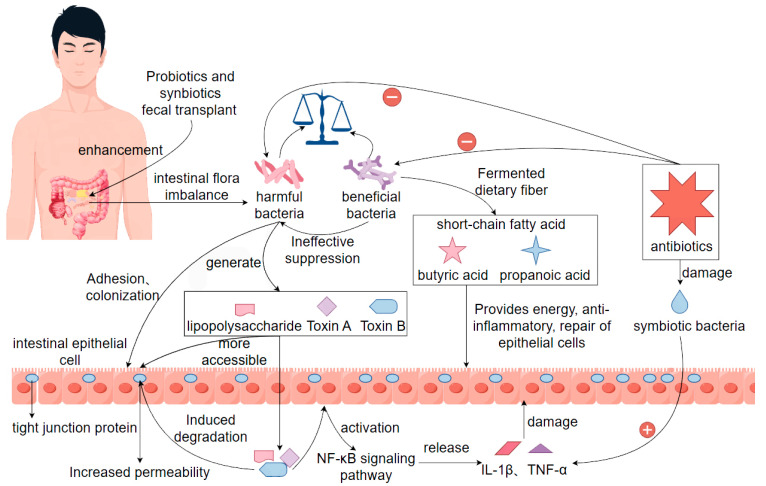
Pathogenesis of bacterial diarrhea due to intestinal flora dysbiosis. Note: ‘+’ indicates promotion or activation; ‘−’ indicates inhibition or suppression.

**Figure 2 biomolecules-15-01045-f002:**
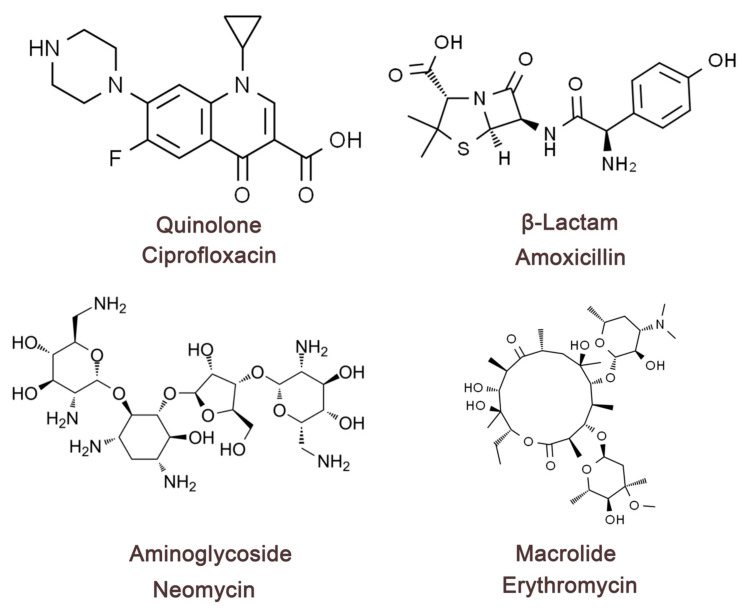
Four types of antibiotic compounds’ chemical structures. Quinolone antibiotics—represented by ciprofloxacin, a fluoroquinolone that inhibits DNA gyrase and topoisomerase IV, thereby disrupting bacterial DNA replication. β-Lactam antibiotics—represented by amoxicillin, a penicillin-type agent that targets penicillin-binding proteins and inhibits bacterial cell wall synthesis. Aminoglycoside antibiotics—represented by neomycin, which binds to the 30S ribosomal subunit and causes mistranslation of bacterial proteins. Macrolide antibiotics—represented by erythromycin, which binds to the 50S ribosomal subunit and inhibits peptide chain elongation.

**Figure 3 biomolecules-15-01045-f003:**
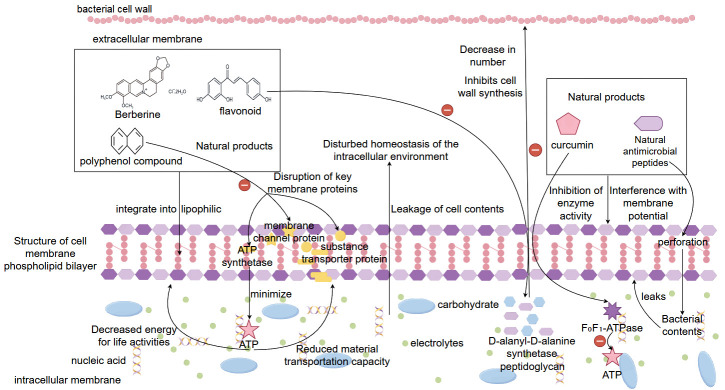
Mechanisms by which natural products treat bacterial diarrhea by interfering with membrane structure, inhibiting membrane lipid synthesis, inducing changes in membrane permeability and membrane dysfunction. Note: ‘+’ indicates promotion or activation; ‘−’ indicates inhibition or suppression.

**Figure 4 biomolecules-15-01045-f004:**
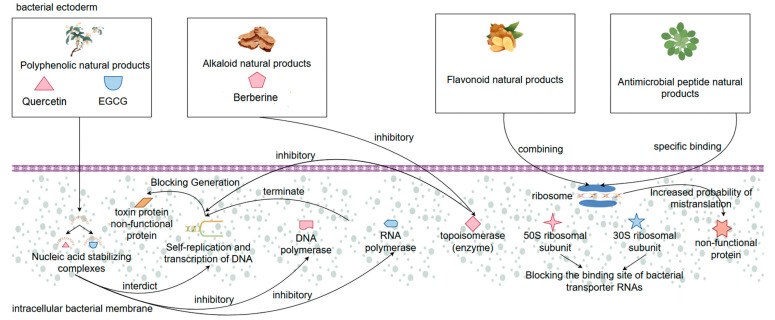
Mechanisms by which natural products treat bacterial diarrhea by disrupting the synthesis of bacterial genetic material and modulating virulence factor expression pathways.

**Figure 5 biomolecules-15-01045-f005:**
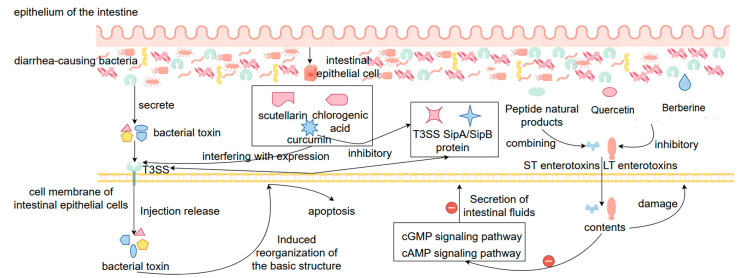
Mechanisms of natural products for the treatment of bacterial diarrhea by inhibiting the synthesis of T3SS and toxins such as LT and ST, toxin secretion, and toxin functions. Note: ‘+’ indicates promotion or activation; ‘−’ indicates inhibition or suppression.

**Figure 6 biomolecules-15-01045-f006:**
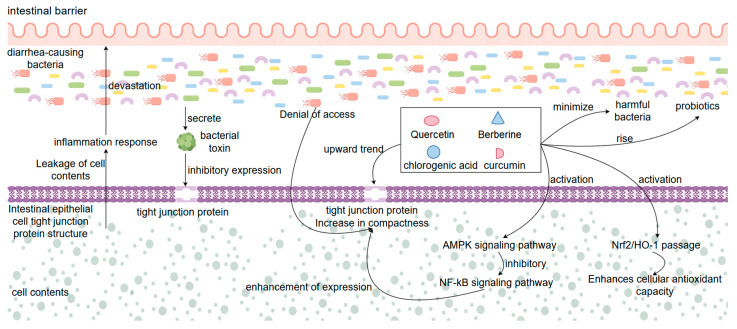
Mechanisms by which natural products treat bacterial diarrhea by protecting intestinal tissues.

**Figure 7 biomolecules-15-01045-f007:**
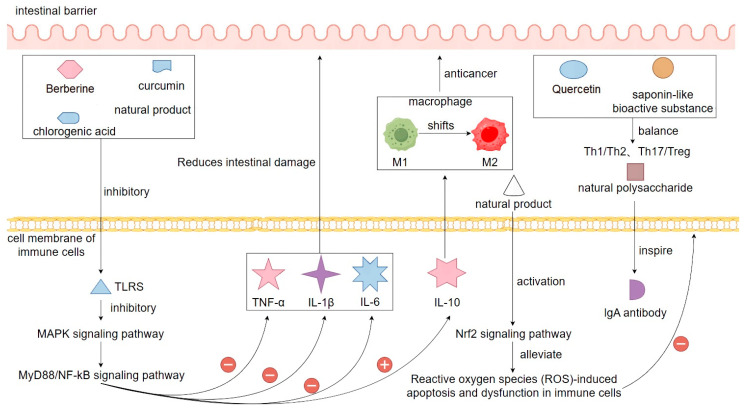
Mechanisms of natural products in the treatment of bacterial diarrhea through immunomodulation. Note: ‘+’ indicates promotion or activation; ‘−’ indicates inhibition or suppression.

**Table 1 biomolecules-15-01045-t001:** Main transmission routes and pathogenic mechanisms of common diarrhea-causing bacteria.

Name of Bacteria	Main Routes of Transmission	Pathogenesis
*Pathogenic* *Escherichia coli*	fecal-oral transmission	Adhesion to intestinal epithelium, destruction of microvilli, formation of adherent adhesion lesions [27].
*Enterotoxigenic Escherichia coli*	fecal-oral transmission	Produces heat-stable toxin (ST) and heat-unstable toxin (LT), which stimulate intestinal secretion and cause watery diarrhea [28].
*Entero-Invasive Escherichia coli*	fecal-oral transmission	Invades intestinal epithelial cells, lyses the cells, and triggers inflammation, leading to mucopurulent bloody stools [29].
*Enterohemorrhagic Escherichia coli*	fecal-oral transmission	Produces Shiga-like toxin, destroying vascular endothelium, causing hemorrhagic colitis, and in severe cases, hemolytic uremic syndrome [30].
*Salmonella*	foodborne transmission	Invades intestinal epithelial cells and induces an inflammatory response, causing diarrhea and systemic symptoms [31].
*Campylobacter*	foodborne transmission	Attacks the intestinal mucosa, produces toxins, destroys the epithelium and causes inflammatory diarrhea [32].
*Shigella*	fecal-oral transmission	Invades colon epithelium, produces Shiga toxin, destroys cells, causes mucopurulent bloody stools [33].
*Clostridium difficile*	fecal-oral transmission	Produces toxins that damage the intestinal epithelium and trigger inflammation, leading to watery diarrhea and pseudomembranous colitis [34].
*Clostridium perfringens*	foodborne transmission	Produces enterotoxins (CPE) that damage the intestinal epithelium, causing diarrhea and enteritis [35].

**Table 2 biomolecules-15-01045-t002:** Representative natural products for the treatment of bacterial diarrhea, their sources, and mechanisms of action.

Name of Natural Product	Molecular Formula	Common Source Plants	Main Mechanism of Action	Mainly Inhibits Bacteria
Baicalein	C_15_H_10_O_5_	Scutellaria baicalensis.	Anti-inflammatory, antimicrobial, suppressor T3SS, *NF-κB* passage [66].	*Salmonella*, *E. coli*, *Shigella*
Quercetin	C_15_H_10_O_7_	Onion, horehound, ginkgo, etc.	Inhibition of toxin expression, antioxidant, immunomodulation [67].	*ETEC*, *EHEC*, *Shigella*
Rutin	C_27_H_30_O_16_	Sophora and mistletoe.	Antioxidant, intestinal barrier protection, anti-inflammatory [68].	*E. coli*, *Salmonella*
Chlorogenic acid	C_16_H_18_O_19_	Honeysuckle, coffee, blueberries, etc.	Anti-inflammatory, intestinal barrier repair, immunomodulation [69].	*Salmonella*, *E. coli*, *Campylobacter*
Catechin	C_15_H_14_O_6_	Tea, grapeseed.	Antimicrobial, antioxidant, membrane interference [70].	*E. coli*, *Shigella*, *V. cholerae*
EGCG	C_22_H_18_O_11_	Green tea.	Inhibition of colonization, biofilm formation, anti-inflammation [71].	*E. coli*, *Salmonella*, *Shigella*
Berberine	C_20_H_18_NO_4_^+^	Huanglian, Phellodendron Bark, Tree Needles.	Antibacterial, anti-inflammatory, regulates intestinal flora, stabilizes barrier [72].	*E. coli*, *Salmonella*, *C. difficile*
Oleanolic acid	C_30_H_48_O_3_	Chasteberry, Hawthorn Leaf.	Anti-inflammatory, antibacterial, enhances mucosal immunity [73].	*E. coli*, *V. cholerae*
Ursolic acid	C_30_H_48_O_3_	Apple Peel, Rosemary, Shiso.	Antibacterial, anti-inflammatory, inhibits toxin expression [74].	*Shigella*, *Salmonella*, *E. coli*
Curcumin	C_21_H_20_O_6_	Turmeric.	Anti-inflammatory, antioxidant, modulation of *TLR/NF-κB* pathway [75].	*Salmonella*, *Shigella*, *Campylobacter*
SCFAs	C_2_H_4_O_2_ C_3_H_6_O_2_ C_4_H_8_O_2_	Gut microbial metabolites.	Regulates intestinal pH, inhibits colonization of pathogenic bacteria, and promotes intestinal barrier repair [76].	*Clostridium difficile*, *E. coli*
Citric acid	C_6_H_8_O_7_	Citrus, fruits, and vegetables.	Lowering intestinal pH, chelating metal ions, indirect bacterial inhibition [77].	broad-spectrum antibacterial
Malic acid	C_4_H_6_O_5_	Apples, hawthorns, etc.	pH lowering, inhibition of bacterial metabolism, synergistic antimicrobial activity [78].	*E. coli*, *Vibrio*

## Data Availability

No new data were created or analyzed in this study.

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
