# Peer review of "A Mechanistic Approach to Replacing Antibiotics with Natural Products in the Treatment of Bacterial Diarrhea"

_biomolecules, 2025, doi:10.3390/biom15071045_

Round 1

Reviewer 1 Report

Comments and Suggestions for Authors

This review addressed how natural products can replace antibiotics in the treatment of bacterial Diarrhea. The manuscript is generally good. However, the following points need to be addressed:

1- The title should be revised to be "A Mechanistic Approach to Replacing Antibiotics with Natural Products in the Treatment of Bacterial Diarrhea" sor a better read.

2- The opening sentence in abstract incorrectly suggests that natural products are already "main drugs" in bacterial diarrhea treatment. If authors can revise the abstract with emphasis on  the natural products as potential alternatives.

-Replace the informal phrase “hot spot” with formal alternatives such as “emerging focus” 

-Explain why exploring natural alternatives is important (e.g., rising antibiotic resistance).

3- The authors should organize the introduction in a logical sequence—starting with the definition and classification of diarrhea, then focusing on bacterial diarrhea and its global prevalence, followed by a discussion of antibiotic use and the rise of resistance, a summary of antibiotic mechanisms of action, and concluding with the mechanisms of bacterial resistance 

-correct taxonomic inaccuracies: Bacillariophyceae are diatoms (algae), not bacteria.

-Use standard bacterial families relevant to human pathogens (e.g., Vibrionaceae, Enterobacteriaceae, Helicobacteraceae).

-I wonder if Oxalis is a plant, not an animal product. Recheck please. The novelty and motivation for this review should be clearly mentioned.

4- (Table 1) should include a column indicating the relevent references. 

5- In page 9-10, structure the text logically by compound classes (Flavonoids, Alkaloids,Terpenoids and organic acids...)

6- Clearly identify mechanistic pathways with  accuracy (e.g., “activating AMPK,” “inhibiting NF-κB,” “stimulating Nrf2/HO-1”), in section 3.4.

7- Elaborate more on the limitations of preclinical animal models in representing human biology, the chemical complexity of natural products that complicates active ingredient identification, issues with bioavailability, insufficient clinical trial data, difficulties in standardization and quality control, and concerns about sustainable sourcing.

8-Emphasize future research directions by highlighting the need for well-designed human clinical trials, advanced extraction and purification techniques, innovative delivery systems to improve bioavailability, and sustainable and ethical sourcing practices to support the scalable production of natural product-based therapies.

9-Ensure all abbreviations, special characters, and formatting are correct and consistent.

10- The entire manuscript should be revised by an English language expert. 

Comments on the Quality of English Language

The entire manuscript should be revised by an English language expert. 

Author Response

Comments 1:

The title should be revised to be "A Mechanistic Approach to Replacing Antibiotics with Natural Products in the Treatment of Bacterial Diarrhea" sor a better read.

Response 1:

Thank you for pointing this out. I agree with this comment. Therefore, I have revised the title accordingly.The updated title is: "A Mechanistic Approach to Replacing Antibiotics with Natural Products in the Treatment of Bacterial Diarrhea".This change can be found on Page 1, Title section of the revised manuscript.

[Updated text in the manuscript]

Title:"A Mechanistic Approach to Replacing Antibiotics with Natural Products in the Treatment of Bacterial Diarrhea".

Comments 2:

The opening sentence in abstract incorrectly suggests that natural products are already "main drugs" in bacterial diarrhea treatment. If authors can revise the abstract with emphasis on  the natural products as potential alternatives.

Replace the informal phrase “hot spot” with formal alternatives such as “emerging focus.

Explain why exploring natural alternatives is important (e.g., rising antibiotic resistance).

Response 2: 

Thank you for pointing this out. I agree with all three suggestions. Therefore, I have made the following changes:

Revised the opening sentence to clarify that natural products are potential alternatives rather than current main drugs.

Replaced the informal phrase “hot spot” with the more formal expression “emerging focus”.

Added a sentence to explain the importance of exploring natural alternatives, specifically in the context of rising antibiotic resistance.

Normative changes were made to the abstract section.

These changes can be found in the revised manuscript on Page 1, Abstract, Lines 14–31.

[Updated text in the manuscript]

Natural products have emerged as potential alternatives to antibiotics in the treatment of bacterial diarrhea, due to their multi-targeting effects, low potential for inducing resistance, and favorable safety profiles. Currently, the search for natural product-based therapies has become an emerging focus in medical research. This growing interest is driven by the increasing awareness that the widespread and irrational use of antibiotics has contributed to the alarming rise of antibiotic-resistant bacterial strains, which in turn diminishes the efficacy of conventional drugs. Among these concerns, the limitations of antibiotics in managing bacterial diarrhea and the potential mechanisms by which natural products exert therapeutic effects are the main focus of this paper. Natural products, containing a wide array of bioactive compounds, can not only directly inhibit the growth of pathogenic bacteria, disrupt bacterial membrane synthesis, and reduce toxin production, but also modulate inflammatory responses, enhance immune function, repair intestinal barriers, and restore gut microbial ecology—highlighting their systemic and multi-targeted therapeutic potential. Therefore, this paper will elaborate on how natural products combat bacterial diarrhea from three aspects: the pathogen and pathogenesis of bacterial diarrhea, natural product-based therapeutic studies, and the underlying mechanisms of action, thereby proposing natural products as viable alternatives to antibiotics.

Comments 3:

The authors should organize the introduction in a logical sequence—starting with the definition and classification of diarrhea, then focusing on bacterial diarrhea and its global prevalence, followed by a discussion of antibiotic use and the rise of resistance, a summary of antibiotic mechanisms of action, and concluding with the mechanisms of bacterial resistance.

correct taxonomic inaccuracies: Bacillariophyceae are diatoms (algae), not bacteria.

Use standard bacterial families relevant to human pathogens (e.g., Vibrionaceae, Enterobacteriaceae, Helicobacteraceae).

I wonder if Oxalis is a plant, not an animal product. Recheck please. The novelty and motivation for this review should be clearly mentioned.

Response 3: 

Thank you for these constructive and detailed suggestions. I have carefully addressed each point as follows:

I reorganized the introduction to follow a more logical and coherent structure. It now begins with the definition and classification of diarrhea, followed by a focus on bacterial diarrhea and its global prevalence, then discusses antibiotic use and the rise of resistance, summarizes antibiotic mechanisms of action, and finally concludes with bacterial resistance mechanisms. This restructuring can be found in the revised manuscript on Pages 2-3, Introduction section, Lines 57–59, Lines 142-144 .

I corrected the taxonomic inaccuracy in the description of bacterial families. Bacillariophyceae, which are diatoms and not bacteria, has been removed and replaced with the correct family Bacillaceae, which is relevant to human bacterial pathogens. This correction appears on Page 1, Introduction section, Line 42 of the revised manuscript.

I have updated the list of bacterial families involved in human intestinal infections to reflect standard nomenclature. The revised text now refers to Vibrionaceae, Enterobacteriaceae, Bacillaceae, and Helicobacteraceae.This can be found on Page 1, Introduction section, Line 42.

Regarding Oxalis, thank you for pointing out the discrepancy. It was a mistake in terminology. The intended animal-derived product is bezoar, not Oxalis. This has been corrected in the manuscript.The revised sentence is on Page 4, Paragraph 1, Line 149.

I added a concise summary at the end of the introduction to clearly highlight the novelty and motivation for this review. The added sentences emphasize the increasing threat of antibiotic resistance and the urgent need for safe, multi-targeted therapeutic alternatives, thus providing context for exploring the role of natural products.These sentences are added on Page 4, Final paragraph of the Introduction, Lines 187-192.

[Updated text in the manuscript]

Vibrionaceae, Enterobacteriaceae, Bacillaceae, and Helicobacteraceae.

With this background, the mechanisms of antibiotics and bacterial resistance have become essential to understand before exploring alternatives.

Against this backdrop, developing safer and more effective alternatives such as natural products has gained increasing attention.

Common natural animal-derived products include propolis, bezoar, and snake venom,

In view of the increasing burden of antibiotic resistance and the clinical challenges of managing bacterial diarrhea, this review aims to comprehensively summarize current knowledge on natural product-based interventions.It highlights their multi-targeted therapeutic mechanisms, safety advantages, and potential for integration into modern anti-infective strategies, thereby providing a scientific basis for the development of antibiotic alternatives.

Comments 4:

(Table 1) should include a column indicating the relevent references.

Response 4:

Thank you for pointing this out. I agree with this comment. Therefore, I would like to clarify that each row in Table 1 already contains its corresponding reference, which has been directly cited within the content of the respective entries. This can be found on Page 5, Lines206-208, Table 1 of the revised manuscript.

[Updated text in the manuscript]

No further changes were necessary, as references are already included in each row of Table 1.

Comments 5:

In page 9-10, structure the text logically by compound classes (Flavonoids, Alkaloids,Terpenoids and organic acids...)

Response 5:

Thank you for pointing this out. I agree with this comment. Therefore, I have revised the section to follow a more logical structure based on compound classes.The revised order is now: flavonoids, alkaloids, terpenoids, organic acids, and polyphenols, which better aligns with conventional classification practices.This change can be found on Pages 13, Lines 482-494.

[Updated text in the manuscript]

Polyphenolic compounds, in addition to encompassing flavonoids, are also widely distributed in natural plant sources such as green tea, grape seeds, and goji berries. Representative bioactive substances, including chlorogenic acid, catechins, and epigallocatechin gallate (EGCG), have been shown not only to exert significant antibacterial activity but also to regulate the gut microbiota, thereby contributing to the maintenance of intestinal homeostasis [95]. Among them, epigallocatechin gallate can not only inhibit the release of toxins by Shigella and reduce its adhesion ability on cells, but also regulate the NF-κB pathway in intestinal cells, thereby reducing the expression of cellular inflammatory factors [96]. Meanwhile, polyphenolic compounds play a role in promoting the growth and development of probiotics and inhibiting the reproduction and expansion of pathogenic bacteria, which play an important role in maintaining the microecological balance of probiotic and pathogenic intestinal microbiota.

References

95. Molinari, R.; Merendino, N.; Costantini, L. Polyphenols as modulators of pre-established gut microbiota dysbiosis:  state-of-the-art.Biofactors. 2022, 48, 255-273.

96. Zhang, Y.; Zhang, Y.; Ma, R.; Sun, W.; Ji, Z. Antibacterial activity of epigallocatechin gallate (egcg) against shigella  flexneri. J. Environ. Res.Public Health2023, 20.

Comments 6:

Clearly identify mechanistic pathways with  accuracy (e.g., “activating AMPK,” “inhibiting NF-κB,” “stimulating Nrf2/HO-1”), in section 3.4.

Response 6:

Thank you for this helpful comment. I agree with this suggestion. Therefore, I have refined the descriptions in Section 3.4 to include more specific mechanistic expressions such as “inhibiting NF-κB,” “activating Nrf2/HO-1,” and “regulating PI3K/Akt”, without altering the original sentence structure. These adjustments help improve the clarity and mechanistic precision of the section.The changes can be found on Pages 11–14, Section 3.4, Lines 410-415, Lines 478-491, Lines 435-438, Lines 524-529, Lines 544-546.

[Updated text in the manuscript]

It has been shown that baicalein has an inhibitory effect on a variety of pathogenic bacteria, such as Salmonella, Shigella, and Escherichia coli, and can inhibit intestinal inflammatory responses and reduce intestinal mucosal damage through the regulation of signaling pathways, such as TLR4/NF-κB (by inhibiting NF-κB activation), MAPK (by suppressing p38 phosphorylation), and so on.

Among them, epigallocatechin gallate can not only inhibit the release of toxins by Shigella and reduce its adhesion ability on cells, but also regulate the NF-κB pathway in intestinal cells (by suppressing NF-κB nuclear translocation), thereby reducing the expression of cellular inflammatory factors.

Among them, curcumin can not only inhibit the apoptosis of intestinal epithelium triggered by pathogenic bacteria, but also promote intestinal repair by regulating signaling pathways such as PI3K/Akt (through activating the PI3K/Akt cascade) and Nrf2/HO-1 (by stimulating Nrf2 nuclear translocation and HO-1 expression).

In addition, the effects of natural products on signaling pathways (e.g., TLR4/NF-κB, MAPK, PI3K/Akt, etc.—including the inhibition of NF-κB activation and activation of PI3K/Akt phosphorylation) and inflammatory factors (e.g., IL-6, TNF-α, and IL-1β) can be observed by fluorescent staining, Western blot, qPCR, ELISA, etc., to further reveal the mechanism of action of the natural drugs.

Curcumin, on the other hand, alleviates oxidative stress and protects intestinal epithelial cells from damage via the Nrf2/HO-1 pathway (by activating Nrf2 and upregulating HO-1 expression).

Comments 7:

Elaborate more on the limitations of preclinical animal models in representing human biology, the chemical complexity of natural products that complicates active ingredient identification, issues with bioavailability, insufficient clinical trial data, difficulties in standardization and quality control, and concerns about sustainable sourcing.

Response 7:

Thank you for this insightful comment. I agree with the concerns raised and have elaborated accordingly in the revised manuscript. In particular, I have supplemented the discussion on the limitations of preclinical animal models, challenges related to chemical complexity, low bioavailability, lack of clinical trial evidence, and difficulties in quality control and standardization. I also added content addressing sustainable sourcing and the need to avoid reliance on endangered plant species.These additions can be found on Pages 23–24, Section 5, Lines 871-903.

[Updated text in the manuscript]

There are notable limitations in current research on the mechanisms by which natural product extracts may replace antibiotics in treating bacterial diarrhea. Most existing studies rely on preclinical animal models, which differ significantly from human biology in terms of immune response, gut microbiota composition, and metabolism. These differences reduce the translational value of animal data and limit the application of findings to human health. Although natural products have demonstrated advantages such as multi-target effects, low potential for resistance, and overall safety, several critical challenges hinder their widespread and rational clinical application.

First, the chemical complexity of natural products—with diverse structural types and multiple bioactive compounds—makes it difficult to isolate the specific active constituents and clearly define their mechanisms of action. Most pharmacological effects arise from the synergistic interaction of these compounds, which also complicates quality control and reproducibility. Many studies still rely on crude extracts or mixtures, lacking purification and quantitative assessment of key active ingredients, which affects batch consistency and therapeutic predictability.Second, bioavailability remains a major bottleneck. Compounds such as polyphenols and flavonoids are often unstable or poorly absorbed when administered orally, as they are rapidly metabolized or degraded in the gastrointestinal tract. This limits the ability to maintain effective drug concentrations at the target site, ultimately compromising clinical efficacy. Novel drug delivery technologies—such as targeted release systems or nanocarriers—are needed to improve absorption and ensure therapeutic levels in vivo.Third, the "slow-acting" nature of many natural antibacterial agents reduces their suitability for acute infections compared to conventional antibiotics, which act rapidly. A potential solution is to explore synergistic treatment strategies, combining natural products with low doses of antibiotics to improve efficacy while minimizing resistance development.Fourth, the lack of large-scale, well-controlled human clinical trials hinders the clinical translation of natural product-based therapies. Most current data are derived from in vitro studies or small animal models, leaving safety, efficacy, dosing, and indications in humans inadequately defined.Fifth, standardization is a major challenge. The composition of natural extracts can vary due to differences in plant origin, harvesting season, and extraction techniques. This variability affects both quality assurance and regulatory approval. Moreover, unsustainable or unethical sourcing—such as overharvesting of endangered species—raises ecological and ethical concerns.

Comments 8:

Emphasize future research directions by highlighting the need for well-designed human clinical trials, advanced extraction and purification techniques, innovative delivery systems to improve bioavailability, and sustainable and ethical sourcing practices to support the scalable production of natural product-based therapies.

Response 8:

Thank you for your helpful suggestion. I have addressed these future research directions in the revised manuscript. Specifically, I have emphasized the importance of multicenter, randomized, controlled clinical trials, modern extraction and purification technologies, novel drug delivery systems to improve bioavailability, and sustainable sourcing strategies to ensure long-term scalability and safety of natural product-based therapies.These additions can be found on Pages 24, Section 5, Lines 904-917.

[Updated text in the manuscript]

In the future, rigorous human clinical studies must be prioritized, alongside the development of advanced extraction and purification technologies to ensure reproducibility and clarity in mechanism. Innovative drug delivery platforms should be employed to enhance bioavailability. Sustainable production should be supported through artificial cultivation, fermentation, or chemical synthesis of bioactive compounds to ensure scalable, ethical sourcing. Furthermore, the integration of multi-omics technologies (e.g., transcriptomics, metabolomics, microbiomics) and computational tools (e.g., network pharmacology) can help elucidate interactions along the host-microbiota-immunity axis.In summary, while natural products present a promising strategy for replacing antibiotics in bacterial diarrhea treatment, their advancement into mainstream clinical application requires overcoming significant scientific, technical, and ethical challenges. Future progress will depend on multidisciplinary collaboration, technological innovation, and policy support to fully realize their therapeutic potential.

Comments 9:

Ensure all abbreviations, special characters, and formatting are correct and consistent.

Response 9:

Thank you for your reminder. I have carefully checked and revised all abbreviations, special characters, and formatting throughout the manuscript to ensure consistency and correctness.

Comments 10:

The entire manuscript should be revised by an English language expert.

Response 10:

Thank you for your suggestion. I have carefully reviewed and revised the entire manuscript to improve grammar, clarity, and readability.

Reviewer 2 Report

Comments and Suggestions for Authors

This manuscript is about the potential of natural products to replace antibiotics in treating bacterial diarrhea. The authors do a good job summarizing the drawbacks of antibiotic use and outlining how natural compounds might offer broader, multi-targeted benefits, including effects on the immune system and gut microbiota. The paper is generally well written and organized. I think it would benefit from a few minor additions to make the discussion more complete and to better highlight its clinical relevance.

  1. One suggestion would be to include short-chain fatty acids (SCFAs) like acetate, propionate, and butyrate as part of the broader discussion on natural products. These microbial metabolites play key roles in maintaining gut homeostasis, regulating inflammation, and enhancing resistance to pathogens. Since they’re endogenously produced but have well-defined bioactive effects, they could reasonably be considered natural products themselves. Including them in the discussion could help round out the mechanistic framework and highlight an additional layer of how natural compounds support gut health. It would also be valuable for the authors to touch on how other bioactive products influence the gut microbiota and SCFA production, further shaping intestinal health.

  1. While I understand that C. difficile infection (CDI) is not the central focus of the manuscript, I suggest briefly addressing it in the discussion as a relevant clinical scenario where natural product-based therapies could be particularly impactful. Since CDI is a well-known consequence of antibiotic-induced dysbiosis, it represents a compelling example of why alternatives to broad-spectrum antibiotics are needed. Including a short paragraph noting whether any of the natural products reviewed (e.g., polyphenols, flavonoids, alkaloids) have shown potential to prevent or ameliorate CDI, either through microbiota modulation, enhancement of colonization resistance, or barrier function, could help illustrate the broader clinical relevance of the field.

Author Response

Comment 1:

One suggestion would be to include short-chain fatty acids (SCFAs) like acetate, propionate, and butyrate as part of the broader discussion on natural products. These microbial metabolites play key roles in maintaining gut homeostasis, regulating inflammation, and enhancing resistance to pathogens. Since they’re endogenously produced but have well-defined bioactive effects, they could reasonably be considered natural products themselves. Including them in the discussion could help round out the mechanistic framework and highlight an additional layer of how natural compounds support gut health. It would also be valuable for the authors to touch on how other bioactive products influence the gut microbiota and SCFA production, further shaping intestinal health.

Response 1:

Thank you for this valuable suggestion. I agree with the reviewer’s comment. Therefore, I have added a comprehensive paragraph discussing the origin, metabolic function, and immunological regulation roles of the three main SCFAs—acetate, propionate, and butyrate—highlighting their physiological and therapeutic relevance to intestinal health. This addition strengthens the mechanistic framework of the manuscript and expands the discussion of how natural products and microbiota-derived metabolites collaboratively support gut health. The revised content can be found on page 12-13, lines 444–478. page 21, lines 788–804.

[Updated Text in the Manuscript]

Acetate is widely produced by bacterial groups such as Bifidobacterium and Bacteroides [87]. With its simple molecular structure, acetate can rapidly cross the intestinal barrier into the bloodstream and be transported to organs such as the liver and muscles, where it plays roles in lipid synthesis, cholesterol metabolism, and appetite regulation. Within the gut, acetate can enhance intestinal epithelial barrier function by activating receptors such as GPR43, promoting the expression of tight junction proteins, and inducing the secretion of mucus and antimicrobial peptides by intestinal epithelial cells, thereby inhibiting the colonization and invasion of pathogenic bacteria [88]. Additionally, acetate can induce the generation of regulatory T (Treg) cells and regulate the Th1/Th2 immune balance, effectively alleviating chronic inflammation caused by dysbiosis [89]. Propionate is primarily produced in the gut through the succinate and propionyl-CoA pathways and serves as a key metabolite linking gut microbiota with host glucose metabolism [90]. In the liver, propionate can act as a substrate for gluconeogenesis and regulate blood glucose homeostasis. Immunological studies have shown that propionate can enhance the chemotaxis and functional maturation of intestinal immune cells by activating receptors such as GPR41, thereby strengthening the mucosal immune barrier. Moreover, propionate can modulate the inflammatory responses of macrophages and dendritic cells, suppressing the overexpression of pro-inflammatory cytokines such as IL-6, IL-1β, and TNF-α. This contributes to alleviating gut inflammation-related disorders such as inflammatory bowel disease (IBD) and irritable bowel syndrome (IBS) [91]. Although butyrate is present in the smallest amount among the major short-chain fatty acids, it has the most critical physiological functions. Butyrate is the primary energy source for colonic epithelial cells, with approximately 70% of it being directly utilized by these cells to promote proliferation and differentiation, thus maintaining the integrity of the intestinal mucosa [92]. Furthermore, butyrate exhibits significant anti-inflammatory effects by inhibiting histone deacetylase (HDAC) activity, affecting inflammatory signaling pathways such as NF-κB, and reducing the expression levels of inflammatory mediators. It also promotes the differentiation of Treg cells and suppresses Th17-related immune responses, thereby maintaining intestinal immune tolerance and reducing the occurrence of autoimmune and chronic inflammatory conditions. Studies have shown that reduced levels of butyrate are closely associated with various intestinal diseases, such as Crohn's disease, ulcerative colitis, and colorectal cancer. Therefore, butyrate is considered an important indicator for assessing gut health and a potential target for therapeutic intervention in intestinal disorders [93]. 

In addition to polysaccharides, various naturally derived bioactive compounds—such as polyphenols, flavonoids, alkaloids, and terpenoids—can also modulate the gut microbiota composition through multiple mechanisms, thereby influencing SCFA production and maintaining intestinal homeostasis. For instance, polyphenolic compounds like tea polyphenols and proanthocyanidins from grape seeds are poorly absorbed in the colon and can be metabolized by specific microbial species. This process promotes the growth of beneficial bacteria (e.g., Bacteroides, Lactobacillus, and Bifidobacterium) while inhibiting opportunistic pathogens, thus optimizing microbial composition [135]. Certain flavonoids can further increase the abundance of butyrate-producing bacteria, elevating overall SCFA levels and enhancing mucosal barrier function and anti-inflammatory responses [136]. Alkaloids such as berberine have been shown to indirectly regulate microbial metabolic activity via the AMPK signaling pathway, leading to increased production of propionate and butyrate [137]. Therefore, these bioactive compounds interact with gut microbiota not only to improve the microbial ecosystem but also to boost SCFA generation, exerting anti-inflammatory, immunoregulatory, and barrier-repairing effects—making them important natural modulators for maintaining and restoring intestinal health.

References

87. Zeng, Z.; Wang, L.; Zhang, W.; Lin, Y.; Wang, B.; Zhang, Y.; Zheng, B.; Pan, L. Elucidation of the pathway for dictyophora indusiata polysaccharide-regulated  differential acetic acid production by bifidobacterium longum f2. Sci. Food. Agric.2025.

88. Zhao, Y.; Chen, F.; Wu, W.; Sun, M.; Bilotta, A.J.; Yao, S.; Xiao, Y.; Huang, X.; Eaves-Pyles, T.D.; Golovko, G.; Fofanov, Y.; D'Souza, W.; Zhao, Q.; Liu, Z.; Cong, Y. Gpr43 mediates microbiota metabolite scfa regulation of antimicrobial peptide  expression in intestinal epithelial cells via activation of mtor and stat3. Mucosal Immunol.2018, 11, 752-762.

89. Liu, X.F.; Shao, J.H.; Liao, Y.T.; Wang, L.N.; Jia, Y.; Dong, P.J.; Liu, Z.Z.; He, D.D.; Li, C.; Zhang, X. Regulation of short-chain fatty acids in the immune system. Immunol.2023, 14, 1186892.

90. Portincasa, P.; Bonfrate, L.; Vacca, M.; De Angelis, M.; Farella, I.; Lanza, E.; Khalil, M.; Wang, D.Q.; Sperandio, M.; Di Ciaula, A. Gut microbiota and short chain fatty acids: implications in glucose homeostasis. J. Mol. Sci.2022, 23.

91. Du Y; He, C.; An, Y.; Huang, Y.; Zhang, H.; Fu, W.; Wang, M.; Shan, Z.; Xie, J.; Yang, Y.; Zhao, B. The role of short chain fatty acids in inflammation and body health. J. Mol. Sci.2024, 25.

92. Gasaly, N.; Hermoso, M.A.; Gotteland, M. Butyrate and the fine-tuning of colonic homeostasis: implication for inflammatory  bowel diseases. J. Mol. Sci.2021, 22.

93. Cui, X.; Li, C.; Zhong, J.; Liu, Y.; Xiao, P.; Liu, C.; Zhao, M.; Yang, W. Gut microbiota - bidirectional modulator: role in inflammatory bowel disease and  colorectal cancer. Immunol.2025, 16, 1523584. 

135. Mithul, A.S.; Wichienchot, S.; Tsao, R.; Ramakrishnan, S.; Chakkaravarthi, S. Role of dietary polyphenols on gut microbiota, their metabolites and health  benefits. Food Res. Int.2021, 142, 110189.

136. Zhou, M.; Ma, J.; Kang, M.; Tang, W.; Xia, S.; Yin, J.; Yin, Y. Flavonoids, gut microbiota, and host lipid metabolism. Life Sci.2024, 24, 2300065.

137. Zhang, X.; Zhao, Y.; Xu, J.; Xue, Z.; Zhang, M.; Pang, X.; Zhang, X.; Zhao, L. Modulation of gut microbiota by berberine and metformin during the treatment of  high-fat diet-induced obesity in rats. Rep.2015, 5, 14405.

Comment 2:

While I understand that C. difficile infection (CDI) is not the central focus of the manuscript, I suggest briefly addressing it in the discussion as a relevant clinical scenario where natural product-based therapies could be particularly impactful. Since CDI is a well-known consequence of antibiotic-induced dysbiosis, it represents a compelling example of why alternatives to broad-spectrum antibiotics are needed. Including a short paragraph noting whether any of the natural products reviewed (e.g., polyphenols, flavonoids, alkaloids) have shown potential to prevent or ameliorate CDI, either through microbiota modulation, enhancement of colonization resistance, or barrier function, could help illustrate the broader clinical relevance of the field.

Response 2:

Thank you very much for your insightful suggestion. We fully agree with your comment. Therefore, we have added a new paragraph to briefly discuss Clostridioides difficile infection (CDI) as a clinically relevant example of antibiotic-induced dysbiosis, in which natural products may hold therapeutic potential. This paragraph summarizes recent findings on polyphenols, flavonoids, and alkaloids that have shown promise in preventing or alleviating CDI by modulating the gut microbiota, enhancing intestinal barrier function, and reducing inflammation or toxin activity. This addition aims to better illustrate the broader clinical relevance of natural product-based therapies as alternatives to antibiotics. The revised content can be found in the Discussion section on Page 9-10, Lines 354–372 of the revised manuscript.

[Updated text in the manuscript ]

In addition, when treating certain types of bacterial diarrhea caused by antibiotic-induced gut microbiota dysbiosis—such as Clostridioides difficile infection (CDI), which is clinically characterized by severe diarrhea, pseudomembranous colitis, and even life-threatening complications—the limitations of current therapies underscore the urgent need for alternatives to broad-spectrum antibiotics. Natural products represent a promising class of such alternatives. Among the wide range of bioactive compounds, several natural substances have shown potential in preventing or alleviating CDI. Polyphenols, such as tea polyphenols and proanthocyanidins from grape seeds, can selectively promote the growth of beneficial bacteria (e.g., Lactobacillus, Bifidobacterium), restore microbial balance, and thereby reduce the risk of C. difficile colonization. Flavonoids like quercetin and baicalin not only exhibit antioxidant and anti-inflammatory properties but also help maintain the expression of tight junction proteins, enhance intestinal barrier integrity, and reduce toxin translocation and inflammation. Alkaloids such as berberine have been demonstrated in several animal models to significantly reduce CDI incidence, with mechanisms including inhibition of toxin production, modulation of bile acid metabolism, and promotion of commensal microbial recovery. These advantages of natural products provide a strong basis for their continued investigation as potential alternatives to broad-spectrum antibiotics in the treatment of bacterial diarrhea.

Reviewer 3 Report

Comments and Suggestions for Authors

Dear authors,

Herein I recommend some suggestions to improve the ease to reach high audience:

  1. In line 125, you write the word "erythema" twice.
  2. Same duplication of word in line 784: "target"
  3.  Lines 238-239, you state: "...induces the large secretion of water and electrolytes, and activates the secretion of water and electrolytes"    It is redundant. Please, correct it.
  4.  In figure 1 and 2 you should explain what does mean the symbols +   and -
  5. In general, figures 2, 3, 4 and 5 are really small. It is not elegant nor clear. You must increase the size of these figures. 
  6. In figure 2, why you include a cell wall?? Cell wall is a characteristic of vegetable cells.
  7.  In figure 3, 4 and 5 you include these sentences: "Japanese empeor oak" and "Obaku school of Zen Buddhism"   What is that?? I am sorry but, why is these sentences written there???
  8.  In subheading 2.3 Mechanisms of action and limitations of antibiotics... I strongly recommend you include a new Scheme where you draw the chemical structures of these compounds, classifying in 4 types of antibiotics.
  9.  Also, in table 2, could you include a new column with the chemical structures of that compounds: Baicalein, Quercetin, Rutin, Chlorogenic acid and so on.
  10. I hope these suggestions improve your good review.
  11. Best regards

Author Response

Comment 1:

In line 125, you write the word "erythema" twice.

Response 1:

Thank you for your careful review. I agree with this comment. Therefore, I have deleted the redundant use of the word "erythema" in line 125 of the revised manuscript.

Comment 2:

Same duplication of word in line 784: "target".

Response 2:

Thank you for pointing this out. I agree with this comment. Therefore, I have deleted the duplicated word “target” in line 784 of the revised manuscript.

Comment 3:

Lines 238–239, you state: "...induces the large secretion of water and electrolytes, and activates the secretion of water and electrolytes." It is redundant. Please, correct it.

Response 3:

Thank you for your careful review. I agree with this comment. Therefore, I have revised the sentence to eliminate redundancy. The corrected sentence now appears in Page 6, Lines 253–255.

[Updated text in the manuscript]:

“…induces the excessive secretion of water and electrolytes into the intestinal lumen, leading to watery diarrhea.”

Comment 4:

In Figure 1 and Figure 2, you should explain what the symbols "+" and "−" mean.

Response 4:

Thank you for your valuable comment. I agree with this suggestion. Therefore, I have added a note to the figure legends of Figure 1 and Figure 2 to clarify the meaning of the symbols. The symbol "+" indicates promotion or activation, and the symbol "−" indicates inhibition or suppression. This revision can be found in the figure legends on Page 8, Page 16 of the revised manuscript.

[Updated figure legend text]:

“Note: ‘+’ indicates promotion or activation; ‘−’ indicates inhibition or suppression.”

Comment 5:

In general, Figures 2, 3, 4, and 5 are really small. It is not elegant nor clear. You must increase the size of these figures.

Response 5:

Thank you for your constructive comment. We fully agree with your suggestion. Therefore, we have adjusted the size and resolution of Figures 2, 3, 4, and 5 to ensure better clarity and visual presentation in the revised manuscript. The updated figures can be found on Page 16, Page 17, Page 18, Page 20.

Comment 6:

In Figure 2, why do you include a cell wall? Cell wall is a characteristic of vegetable (plant) cells.

Response 6:

Thank you for your comment. We appreciate your attention to detail. We would like to clarify that the inclusion of the “cell wall” in Figure 2 refers to the bacterial cell wall, which is a well-established structural component of bacterial cells (e.g., peptidoglycan layer in Gram-positive and Gram-negative bacteria). However, to avoid confusion with plant cell walls, we have revised the label in Figure 2 to specify “bacterial cell wall” . This correction can be found on Page 16 of the revised manuscript.

Comment 7:

In Figures 3, 4, and 5 you include the sentences: "Japanese emperor oak" and "Obaku school of Zen Buddhism." What is that? I am sorry, but why are these sentences written there?

Response 7:

Thank you for pointing out this issue. We sincerely apologize for the confusion. This was caused by a language translation error. Specifically, "Japanese emperor oak" was incorrectly used in place of “quercetin,” and "Obaku school of Zen Buddhism" was mistakenly inserted instead of “berberine.” These phrases were generated due to an automatic translation artifact during figure preparation.

We have corrected these labels in Figures 3, 4, and 5 to accurately reflect the intended compounds, “quercetin” and “berberine,” and have reviewed all figures to ensure no other unintended labels remain. The corrected versions can be found on Page 17, Page 18, Page 20.

We appreciate your careful review and sincerely regret the oversight.

Comment 8:

In subheading 2.3 "Mechanisms of action and limitations of antibiotics," I strongly recommend you include a new scheme where you draw the chemical structures of these compounds, classifying them into four types of antibiotics.

Response 8:

Thank you very much for your insightful suggestion. I agree with this recommendation. Therefore, I have included a new Scheme 1 in Section 2.3 (Page 9) of the revised manuscript. This scheme illustrates the representative chemical structures of four major antibiotic classes: Quinolones, β-Lactams, Aminoglycosides, and Macrolides.

Comment 9:

Also, in Table 2, could you include a new column with the chemical structures of the compounds: Baicalein, Quercetin, Rutin, Chlorogenic acid, and so on.

Response 9:

Thank you for your helpful suggestion. I agree with your comment. Therefore, I have revised Table 2 by adding a new column presenting the chemical structures of the representative natural compounds discussed, including Baicalein, Quercetin, Rutin, Chlorogenic acid, and others. This visual addition enhances the reader’s understanding of the molecular features of these bioactive substances. The updated Table 2 can be found on Page 10, Lines 387-388 of the revised manuscript.

Comment 10:

I hope these suggestions improve your good review.

Response 10:

Thank you very much for your positive and constructive feedback. I sincerely appreciate your valuable suggestions, which have greatly helped improve the clarity, accuracy, and overall quality of this review. I have carefully addressed all the comments and made the corresponding revisions in the manuscript accordingly.

Comment 11:

Best regards

Response 11:

Best regards

Round 2

Reviewer 1 Report

Comments and Suggestions for Authors

The manuscript has been improved significantly to the satisfaction of the reviewer. 

Author Response

Comments 1:

The manuscript has been improved significantly to the satisfaction of the reviewer.

Response 1:

I sincerely thank the reviewer for the positive and encouraging feedback. I am very pleased to know that the revised manuscript has been improved to the reviewer’s satisfaction. I truly appreciate the valuable comments and suggestions provided during the review process, which greatly helped me refine the scientific quality and clarity of the work.

Please do not hesitate to let me know if any additional revisions are needed.

Reviewer 2 Report

Comments and Suggestions for Authors

The authors have revised the manuscript and incorporated my comments; the current version of the paper is suitable for publication.

Author Response

Comments 1:

The authors have revised the manuscript and incorporated my comments; the current version of the paper is suitable for publication..

Response 1:

I sincerely thank the reviewer for the positive evaluation. I am pleased to know that the revised manuscript has addressed all the comments and is now considered suitable for publication. I truly appreciate the reviewer’s constructive suggestions and time spent during the review process, which greatly contributed to improving the quality of this work.

Reviewer 3 Report

Comments and Suggestions for Authors

Dear authors,

Many thanks for the introduction of all my suggestions. I honestly think that the manuscript is now quite better. Please, just revise now the new figure 2, where you have some wrong chemical structures: Beta-lactam structure is wrong; Macrolide chemical structure is also wrong. Pay attention to these important fails. Please, correct them!

Best regards

Author Response

Comments 1:

Many thanks for the introduction of all my suggestions. I honestly think that the manuscript is now quite better. Please, just revise now the new figure 2, where you have some wrong chemical structures: Beta-lactam structure is wrong; Macrolide chemical structure is also wrong. Pay attention to these important fails. Please, correct them!

Best regards

Response 1:

Thank you very much for your careful review and constructive suggestions. I fully agree with your comment. Accordingly, I have carefully revised the chemical structures in Figure 2, specifically correcting the β-lactam structure (amoxicillin) and the macrolide structure (erythromycin). The updated structures have been rechecked and redrawn based on verified chemical databases (e.g., PubChem), ensuring their accuracy.

In addition, I have revised the figure legend for clarity and accuracy. The updated figure and legend now appear on page 17, Figure 2, and the legend is modified as follows:

[Updated text in the manuscript]

Figure 2. Four types of antibiotic compounds' chemical structures.

Quinolone antibiotics—represented by ciprofloxacin, a fluoroquinolone that inhibits DNA gyrase and topoisomerase IV, thereby disrupting bacterial DNA replication.

β-Lactam antibiotics—represented by amoxicillin, a penicillin-type agent that targets penicillin-binding proteins and inhibits bacterial cell wall synthesis.

Aminoglycoside antibiotics—represented by neomycin, which binds to the 30S ribosomal subunit and causes mistranslation of bacterial proteins.

Macrolide antibiotics—represented by erythromycin, which binds to the 50S ribosomal subunit and inhibits peptide chain elongation.

I appreciate your attention to detail, which has helped improve the scientific accuracy and overall quality of the manuscript.